bioengineering/biomechanics

flapping wings, wind gusts, vertical inflows, disturbances, wake interaction

**Author for correspondence:**
Soudeh Mazharmanesh
e-mail: soudeh.mazharmanesh@.adfa.edu.au

# Effects of uniform vertical inflow perturbations on the performance of flapping wings

Soudeh Mazharmanesh[1], Jace Stallard[1], Albert Medina[2], Alex Fisher[3], Noriyasu Ando[4], Fang-Bao Tian[1], John Young[1] and Sridhar Ravi[1]

[1]School of Engineering and Information Technology, University of New South Wales, Canberra, ACT 2600, Australia
[2]US Air Force Research Laboratory, Wright-Patterson Air Force Base, OH 45433, USA
[3]School of Engineering, RMIT University, Melbourne, 3083, Australia
[4]Department of System Life Engineering, Maebashi Institute of Technology, Maebashi 371-0816, Japan

SM, 0000-0002-3263-760X; JY, 0000-0002-8671-990X;
SR, 0000-0001-7397-9713

Flapping wings have attracted significant interest for use in miniature unmanned flying vehicles. Although numerous studies have investigated the performance of flapping wings under quiescent conditions, effects of freestream disturbances on their performance remain under-explored. In this study, we experimentally investigated the effects of uniform vertical inflows on flapping wings using a Reynolds-scaled apparatus operating in water at Reynolds number $\approx 3600$. The overall lift and drag produced by a flapping wing were measured by varying the magnitude of inflow perturbation from $J_{\text{Vert}} = -1$ (downward inflow) to $J_{\text{Vert}} = 1$ (upward inflow), where $J_{\text{Vert}}$ is the ratio of the inflow velocity to the wing's velocity. The interaction between flapping wing and downward-oriented inflows resulted in a steady linear reduction in mean lift and drag coefficients, $\bar{C}_L$ and $\bar{C}_D$, with increasing inflow magnitude. While a steady linear increase in $\bar{C}_L$ and $\bar{C}_D$ was noted for upward-oriented inflows between $0 < J_{\text{Vert}} < 0.3$ and $J_{\text{Vert}} > 0.7$, a significant unsteady wing–wake interaction occurred when $0.3 \leq J_{\text{Vert}} < 0.7$, which caused large variations in instantaneous forces over the wing and led to a reduction in mean performance. These findings highlight asymmetrical effects of vertically oriented perturbations on the performance of flapping wings and pave the way for development of suitable control strategies.

# 1. Introduction

Remarkable agility and manoeuvrability of insects have led to an increasing interest in insect-inspired flight mechanisms for small-scaled flying vehicles (Reynolds number, $Re < 10\,000$) [1–3]. At the same time, bioinspired miniature robots with flapping wings have been also found to be useful tools in exploring the mechanics of insect flight [4]. Flapping wings offer many potential advantages for small-scaled engineered flying vehicles, such as high manoeuvrability and relatively low noise than rotary wings [5]. The successful implementation of such small-scale vehicles strongly depends on their ability to operate efficiently in wind disturbances, likely to be encountered during the flight in cluttered environments such as in proximity to buildings, under tree canopies, or in mountainous terrain with updrafts and downdrafts [6]. The response of flapping wings to freestream disturbances is expected to be fundamentally different to that of fixed or rotary wings due to their inherently unsteady flow profile over the wing, and the constantly varying wing velocity and orientation. Analysis of the effects of aerodynamic perturbations on the performance of flapping wings is thus a critical component in the construction of miniature unmanned flying vehicles that can operate effectively in realistic environments (table 1).

In this context, the interaction between flapping wings and freestream disturbances may be characterized as either fast or slow, depending on the temporal profile of the disturbance in relation to the wingbeat. Fast disturbances (or 'gusts') are those where the duration over which the wing experiences a significant portion of the disturbance magnitude (as a fraction of the wing velocity) is of the same order or smaller than the timescale of their wingbeat, while slow disturbances (or 'inflows') may be considered as those with frequencies of variation much lower than the wing flapping frequency. Due to the high wingbeat frequencies of small size-scaled flapping wing flyers, it is likely that the majority of the perturbations encountered in realistic environments are likely to be of timescales much lower than those of the wingbeat. Both gusts and inflows are considered in the flapping wing literature, although the term 'gust' has been used interchangeably to describe both fast and slow disturbances.

While the interaction between freestream inflows (perturbations) and flapping wings can occur in different orientations, most studies available in the literature have investigated the effects of frontally and laterally oriented inflows using numerical [7–9] and experimental [10–13] techniques. These studies typically employed computational fluid dynamics or experimental flow visualization to quantify instantaneous flow around the flapping wings in the presence of lateral and frontal inflows and evaluated corresponding aerodynamic properties such as instantaneous and time-averaged lift and drag coefficients. Frontal and lateral inflows contributed to increase lift, whereas drag was found to be linearly proportional to flow speed [9,12,13]. While Chirarattananon *et al.* [14] experimentally showed that the effects of frontal inflows were more pronounced compared with oriented laterally, Jones & Yamaleev [9] revealed that flapping wings can alleviate the effect of moderate or even strong frontal inflows whose mean velocity is comparable with the wing tip velocity. In addition, their numerical results revealed that the mean lift tended to increase when the wing oriented towards the lateral inflow, whereas it decreased when the wing was oriented away from the lateral inflow.

There are few studies which have specifically focused on the interaction between vertically oriented inflows and flapping wings [15,16]. The effects of downward-oriented inflows on lift and drag forces were explored numerically by Jones & Yamaleev [9]. They revealed that a downward inflow reduced the time-averaged lift force compared with the case with quiescent flow due to reduction in the effective angle of attack which led to a smaller leading-edge vortex (LEV) to form over the wing compared with quiescent flow.

Recently, Jakobi *et al.* [17] examined bees flying through flows of different orientations and noted that bees were more perturbed by downward as compared with upward flows. Motivated by the application of flapping wings outside of laboratory conditions and inspired by the gust response of insects in nature; in the present work, we consider the interaction between flapping wings and inflows oriented upward or downward with respect to the stroke plane. Most vertical flow studies available in the literature have been performed numerically for two-dimensional wings, assuming symmetry over the span and mainly focused on downwards inflows with low velocities ($J_{\mathrm{Vert}} < 0.5$).

Here, we experimentally tested the aerodynamic performance of a flapping wing while introducing a range of vertically oriented flows of constant velocity (perpendicular to the stroke plane). Force measurements were taken through a six-axis force/torque (F/T) sensor to gain an understanding of the effect of the vertical flow on the mean and transient aerodynamic performance of the wing. Flow behaviour was also captured through digital particle image velocimetry (DPIV) to identify key flow features and to measure the effective angle of attack of the incoming flow for a range of upwards and downwards vertical inflows.

**Table 1.** Nomenclature.

| | | | |
|---|---|---|---|
| $C_L$ | coefficient of lift $\left(2F_{y_G}/\rho U_{RoG}^2 S\right)$ | $b$ | wingspan [m] |
| $C_D$ | coefficient of drag $\left(2F_{x_G}/\rho U_{RoG}^2 S\right)$ | $c$ | wing chord [m] |
| $\bar{C}_L$ | mean coefficient of lift | $f$ | flapping frequency [Hz] |
| $\bar{C}_D$ | mean coefficient of drag | $v$ | flow vertical velocity component [m s$^{-1}$] |
| $F_{x_L}$ | wing normal force [N] | $\mathbf{v}$ | flow velocity vector [m s$^{-1}$] |
| $F_{y_L}$ | wing tangential force [N] | $w$ | flow horizontal velocity component [m s$^{-1}$] |
| $F_{x_G}$ | drag force [N] | $\theta$ | pitch angle [°] |
| $F_{x_{G'}}$ | drag force normal to wingspan [N] | $\theta_{max}$ | maximum pitch angle [°] |
| $F_{y_G}$ | lift force [N] | $\Theta_{eff}$ | effective angle of attack [°] |
| $I$ | second moment of area [m$^4$] | $\Theta_{th}$ | theoretical angle of attack [°] |
| $J_{Vert}$ | vertical inflow ratio ($U_{inflow}/U_{RoG}$) | $\mu$ | fluid viscosity [N s m$^{-2}$] |
| $Q$ | $Q$-criterion [s$^{-2}$] | $\rho$ | density [kg m$^{-3}$] |
| $Re$ | Reynolds number ($\rho U_{RoG}c/\mu$) | $\sigma$ | standard deviation |
| $R_{RoG}$ | wing's radius of gyration [m] | $\bar{\sigma}$ | mean standard deviation |
| $R_{off}$ | wing offset [m] | $\phi$ | stroke position [°] |
| $S$ | wing surface area [m$^2$] | $\phi_{max}$ | maximum flapping amplitude [°] |
| $T$ | flapping period of wing beat [s] | $[\psi]$ | strain rate tensor [1 s$^{-1}$] |
| $U_{RoG}$ | wing radial velocity at radius of gyration [m s$^{-1}$] | $\omega$ | vorticity [1 s$^{-1}$] |
| $U_{inflow}$ | inflow velocity [m s$^{-1}$] | $[\Omega]$ | rotation rate tensor [1 s$^{-1}$] |

# 2. Experimental set-up and procedure

## 2.1. Flapping wing rig

The experiment consisted of a dynamically scaled, single flapping wing operated in a $900 \times 900 \times 600$ mm water tank (figure 1a). The wing was rectangular with 34 mm chord and 102 mm span (table 2), resulting in over four wingspans separation between the wingtip and the tank walls at all times. The flapping rig itself consisted of two servo motors (RoboStar SBRS-5314HTG 280°, Digital Gear High Voltage Robot Servo) for wing kinematic control. One servomotor controlled the flapping motion of the wing via the main shaft while another servomotor controlled the pitching motion of the wing through a pulley system, figure 1a. A six-axis force/torque sensor (ATI Nano 17-IP68) was mounted between the main shaft of the flapper and the root of the wing. The flapping rig was attached to a vertical linear actuator (multi-axis ball screw linear motion stage, FUYU Motion) consisting of a stepper-motor (NEMA 23, STEPPERONLINE) controlled leadscrew which allowed for 400 mm of vertical motion to 0.05 mm position accuracy.

Vertical inflows were introduced by translating the flapping rig along a vertical rail that was capable of implementing arbitrary upwards and downwards motions. The flapping rig was supported by a vertical guide and wheel (figure 1a) which stabilized the system during vertical motion. The flow profile over the wing was quantified by two-dimensional DPIV, with imagery being collected by a high-speed camera (CR3000 × 2, Optronis) placed perpendicular to the light sheet to capture the flow field.

The wing was constructed using a rigid 1 mm plastic sheet as shown in figure 1b. In order to accommodate the F/T sensor, a rectangular cut out was included in the top corner of the wing. There was also an offset from the flapping axis of rotation to the root of the wing of 20 mm to accommodate flapping rig attachment mechanisms (figure 1b).

## 2.2. Wing kinematics and test procedure

The kinematics selected for this study (table 2, figures 2 and 3) involved sinusoidal flapping in a horizontal stroke plane with amplitude $\phi_{max}$ of 120° (measured from the aft-most position of the wing at the start of the

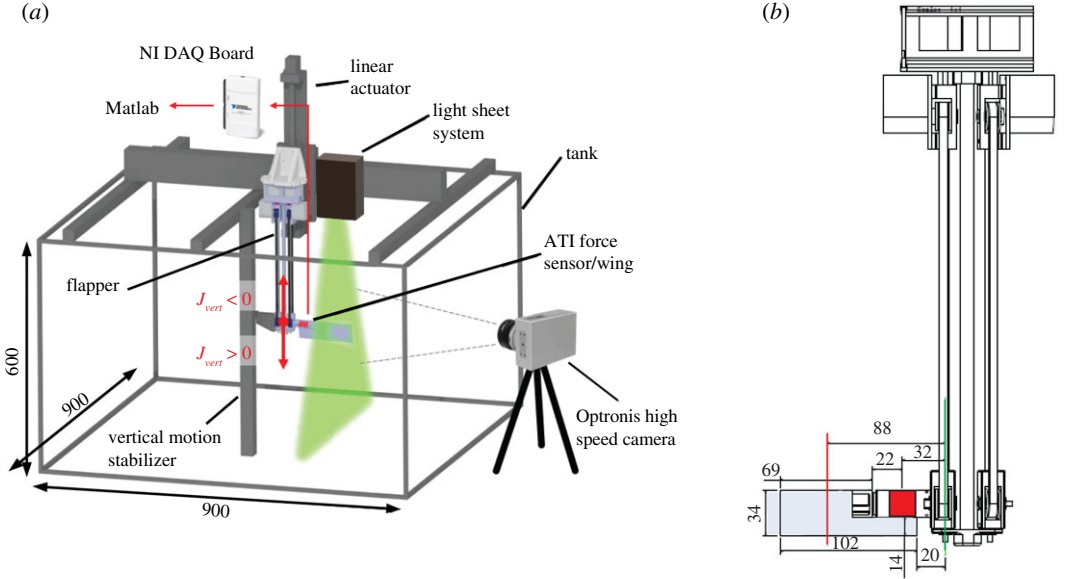

**Figure 1.** (*a*) Schematic illustration of the experimental set-up. The test set-up can be seen with the flapping rig mounted onto a vertical linear actuator and extending into the water tank. The wing and F/T sensor are mounted at the bottom of the rig opposite the guide connected to the vertical stabilizer for stability. The light sheet and camera for DPIV are also shown in their respective positions. (*b*) Front view of assembled flapper, wing and force sensor schematics. The wing and force sensor are highlighted in blue and red, respectively; the axis of rotation being the vertical green line and the wing radius of gyration $R_{RoG}$ indicated by the vertical red line. All dimensions are in mm.

**Table 2.** Experiment kinematics, properties and dimensions.

| experiment property | value |
|---|---|
| flapping frequency ($f$) | 0.254 Hz |
| flapping amplitude ($\phi_{max}$) | 120° |
| pitching angle ($\theta_{max}$) | 45° |
| pronation and supination duration | 20% of wingbeat |
| wingspan ($b$) | 102 mm |
| chord ($c$) | 34 mm |
| wing offset ($R_{off}$) | 20 mm |
| radius of gyration $\left(R_{RoG} = \sqrt{I/S}\right)$ | 88 mm |
| Reynolds number ($Re = \rho U_{RoG} c/\mu$) | 3600 |

forward stroke). Non-sinusoidal pitching kinematics were used, with each forward stroke and back stroke consisting of pronation and supination phases (the first and second rotation phases of the stroke, respectively each with duration 20% of the stroke) and a translation phase with constant pitch angle ($\theta_{max}$) of 45° (measured from the vertical as shown in figure 3). These parameters nominally match the flapping kinematics of bees and wasps [18]. The flapping and pitching motions of forward and back strokes were symmetric. In what follows, 'stroke' refers equally to both forward and back strokes.

The radius of gyration $\left(R_{RoG} = \sqrt{I/S}\right)$ was selected as the reference length, where $I$ and $S$ are the second moment of area and surface area of the wing, respectively, similar to previous studies [19–22]. The velocity at the radius of gyration ($U_{RoG} = 2\phi_{max} f R_{RoG}$) was used as the reference velocity and a $Re = \rho U_{RoG} c/\mu$ of 3600 based on $U_{RoG}$ and chord length was selected and kept constant in the present study. At this Reynolds number, representative of bees (*Bombus lapidarius* $Re = 3700$) and moths (*Pieris brassicae* $Re = 4000$) [18], LEVs play an important role in the generation of aerodynamic forces [20]. It may be hypothesized that vertical inflows will significantly interact with the formation and evolution of the LEV throughout the wingbeat by altering the effective angle of attack time history.

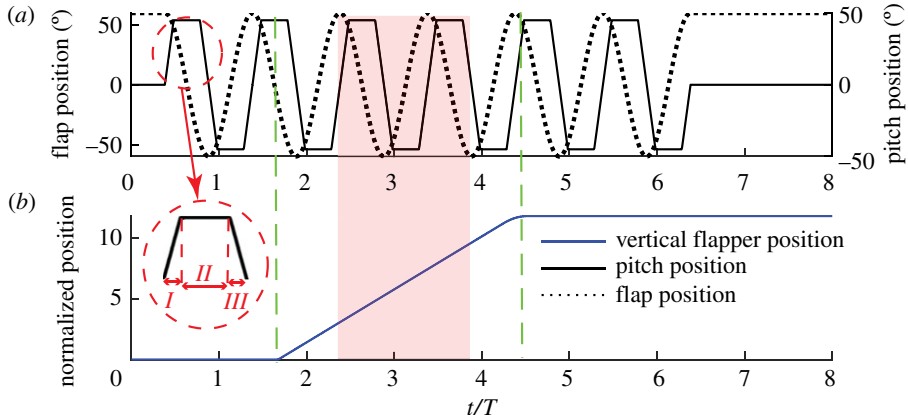

**Figure 2.** Flapping kinematic waveform. The flapping kinematics for an example test are shown here. The dashed line displays the sinusoidal flapping wave form, and the solid line shows the constant rotational velocity during pitch rotation. Strokes which were recorded are highlighted in red. The inset shows the stages of the stroke labelled with pronation (*I*), translation (*II*) and supination (*III*). *T* is the flapping period of the complete wing beat. In (*b*), y-axis was normalized using chord length.

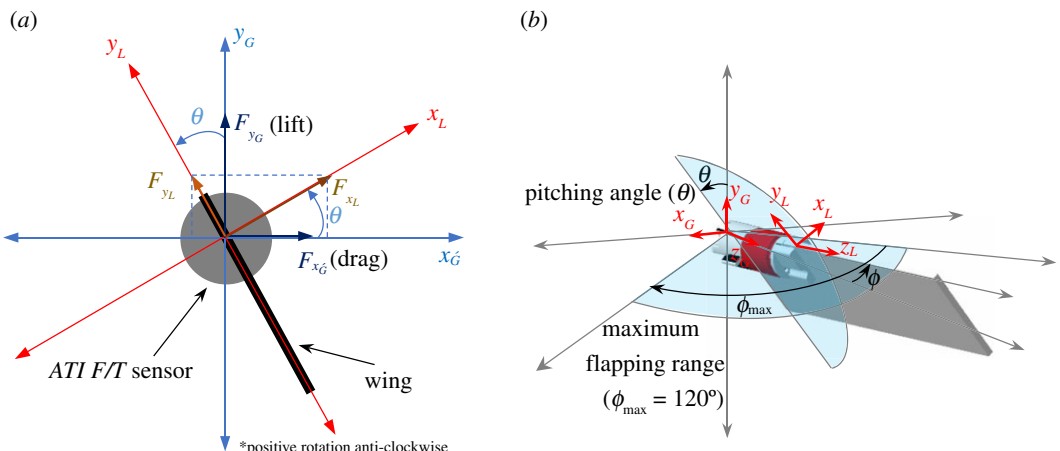

**Figure 3.** Coordinate systems. (*a*) Outlines the body-fixed (*L*) and inertial (*G*) coordinate system's relation to pitch angle. (*b*) The body-fixed and inertial coordinate systems given the pitching and flapping angle of the wing.

Here, we describe the experiment procedure followed for the upward-oriented inflow perturbation, noting that the same procedure with opposite inflow direction was followed for downward inflow conditions. The wing was initially flapped in quiescent fluid (i.e. in pure hover), at a distance of three chord lengths from the free surface, at the top of the experiment tank, where surface effects were negligible. No disturbance of the water surface was observed. To introduce an upwards inflow, the flapping rig was translated downwards over a distance of 400 mm ($\approx$12 chord lengths) with short acceleration and deceleration phases and constant velocity in between (dashed green lines in figure 2). The duration of the acceleration and deceleration phases accounted for 1% of the entire duration of the inflow. To reduce both the transient effect from the acceleration phase at the start of the inflow, and any interference with the wake produced during the initial pure hover before inflow commencement, force measurements were commenced only after one complete stroke following the start of the inflow. Data were then collected for all subsequent strokes prior to commencement of the deceleration phase (red shading in figure 2). After deceleration, the induced inflow was stopped at three chord lengths above the bottom of the tank, thereby minimizing the ground effects at the end of the motion (figure 2). The same procedure was employed for the downward inflow perturbations, with upwards translation of the rig from three chord lengths from the bottom of the tank, to approximately three chord lengths below the free surface.

Variation in vertical inflows was non-dimensionally represented as an inflow ratio, $J_{\text{Vert}}$, which was the ratio of the inflow velocity to $U_{\text{RoG}}$ as shown in equation (2.1).

$$J_{\text{Vert}} = \frac{U_{\text{inflow}}}{U_{\text{RoG}}} = \frac{U_{\text{inflow}}}{2\phi_{\max} f R_{\text{RoG}}}, \tag{2.1}$$

$J_{\text{Vert}} < 0$ denote downwards inflows while $J_{\text{Vert}} > 0$ denote upwards inflows. The maximum range of inflow ratios analysed in this study was $J_{\text{Vert}} = \pm 1$, which corresponds to the velocity of the vertical flow equal to $U_{\text{RoG}}$. For bees and wasps that operate at approximately $Re = 3600$, during hover, $J_{\text{Vert}} = \pm 1$ indicates an upward or downward wind velocity of approximately 5–6 m s$^{-1}$ (based on the wing tip velocity of these insects). This range of $J_{\text{Vert}}$ is also representative of the range of winds and vertical flows experienced in the outdoor environment [23].

## 2.3. Force measurement and data acquisition

Force measurements were collected using an ATI Nano-17 IP68 force/torque (F/T) sensor which was mounted at the root of the wing. The sensor was connected to a signal conditioner and a PCI-6143 National Instruments DAQ Board linked to a PC. Data were processed using in-house-developed code in Matlab. The F/T sensor measured $F_{x_L}$ and $F_{y_L}$ corresponding to the wing's body-fixed normal and tangential forces, respectively, as displayed in figure 3a.

The raw force collected by the F/T sensor comprised the weight of the wing/sensor (including its buoyancy force), the inertia force from wing/sensor motion and the fluid mechanic force. The barometric pressure of the fluid during vertical motion had to be accounted for; however, this was linearly proportional to depth and was removed by applying a correction which was calculated based on the maximum pressure difference over 400 mm vertical translation, the instantaneous inflow velocity and the corresponding depth. A single weight measurement resolved into body-fixed coordinates would potentially introduce an error when subtracting from the other measured forces due measurement uncertainty in the instantaneous wing pitch angle. Accordingly, the weight force (including buoyancy) was directly measured in body-fixed coordinates $x_L$ and $y_L$ (figure 3a) at different $\theta$, by rotating the wing very slowly in water to ensure no fluid mechanic or inertial forces affected the weight readings. Coupled with the pitch kinematics of the wing, the weight force (including buoyancy) could then be subtracted from the raw force data. The inertial force was measured by flapping the wing with the same test kinematics in air. Due to the considerably lower density of air compared with water the wing experienced negligible fluid mechanic forces, isolating inertial and weight force. The weight force in air was measured and removed using the same method as in water, leaving only the inertial force which was phase averaged over 40 strokes to give an average inertial force in body-fixed coordinates for a single stroke. The fluid mechanic force was isolated by removing weight and inertia from the raw F/T sensor readings through equation (2.2). The force components $F_{y_G}$ (lift) and $F_{x_G}$ (drag) in inertial coordinates $x_G$ and $y_G$ (figure 3b) were calculated using the pitching and flapping positions $\theta$ and $\phi$ through equations (2.3) and (2.4). The data were phase averaged over 20 strokes and filtered using a low-pass, fourth-order, Butterworth filter with a cut-off frequency of 8 Hz which was significantly higher than the flapping frequency of the wing (0.254 Hz). A similar method was used by Nagai et al. [24] and Bhat et al. [20] to isolate the fluid mechanic forces in a Reynolds-scaled apparatus.

$$\left.\begin{aligned} F_{x_L, y_L \,(\text{sensor})} &= F_{x_L, y_L \,(\text{fluid mechanic})} + F_{x_L, y_L \,(\text{weight})} + F_{x_L, y_L \,(\text{inertia})} \\ F_{x_L, y_L \,(\text{fluid mechanic})} &= F_{x_L, y_L \,(\text{sensor})} - F_{x_L, y_L \,(\text{weight})} - F_{x_L, y_L \,(\text{inertia})}, \end{aligned}\right\} \tag{2.2}$$

and

$$\begin{bmatrix} F_{y_G} \\ F_{x_{G'}} \end{bmatrix} = \begin{bmatrix} \sin(\theta) & \cos(\theta) \\ \cos(\theta) & -\sin(\theta) \end{bmatrix} \begin{bmatrix} F_{y_L} \\ F_{x_L} \end{bmatrix} \tag{2.3}$$

and

$$F_{x_G} = F_x G' \cos\phi. \tag{2.4}$$

For the selected kinematics and Reynolds number, the fluid mechanic forces were found to be symmetric for the forward and back strokes, and in the following analysis, the results for both are considered together. Following Han et al. [13], the instantaneous lift and drag coefficients $C_L$ and $C_D$ were calculated according to

$$C_L = \frac{2F_{y_G}}{\rho U_{\text{RoG}}^2 S} \quad \text{and} \quad C_D = \frac{2F_{x_G}}{\rho U_{\text{RoG}}^2 S}. \tag{2.5}$$

The surface area $S$ of the wing was the total area of the wing excluding the cut-out for the force sensor (figure 1b). The density of the fluid $\rho$ was 1000 kg m$^{-3}$. Mean lift and drag coefficients ($\bar{C}_L$ and $\bar{C}_D$) were obtained by time-averaging the phase-averaged forces and indicate averages over a single stroke rather than the entire wing beat.

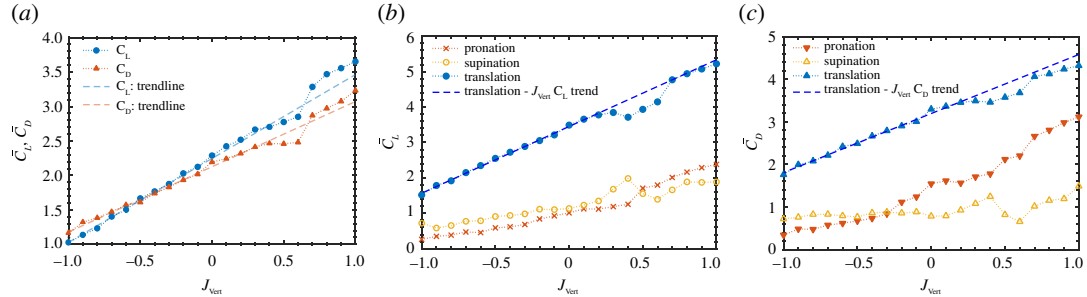

**Figure 4.** Overall and stroke components values of $\bar{C}_L$ and $\bar{C}_D$. (*a*) Mean overall lift and drag coefficients $\bar{C}_L$ and $\bar{C}_D$ versus $J_{\text{Vert}}$. (*b*) Mean lift $\bar{C}_L$ for three phases of the stroke. (*c*) Mean drag $\bar{C}_D$ for three phases of the stroke.

## 2.4. Particle image velocimetry

The flow field over the wing was visualized using two-dimensional DPIV captured through a high-speed camera (CR3000 × 2, Optronis) operating at 300 frames per second with a shutter speed of 1/300 s and resolution 1696 × 1710 pixels observing a light sheet (4 mm thickness) produced by an ultra-bright light-emitting diode (LED; LPS3, iLA5150GmbH) (figure 1*a*). Ten micrometres of neutrally buoyant hollow glass spheres were used to highlight flow motion within the light sheet. The camera was controlled by a trigger pulse generated by TimeBench analysis software. All image pairs were captured perpendicular to the glass sides of the tank to remove refraction effects.

The images were captured at the centre of the translation phase (middle of the stroke) with the light sheet positioned at the $R_{\text{RoG}}$. Mid-stroke and $R_{\text{RoG}}$ were selected for DPIV as the LEV could be clearly observed at this location, without LEV breakdown [25] or bifurcation [20]. Due to the presence of strong chordwise flow, most of the particle motion was along the illuminated plane, and the effects of out-of-plane motion were lessened by reducing image exposure during frame acquisition, still ensuring an acceptable signal-to-noise ratio. A 55 × 55 mm window was used to capture the images. A pair of images was collected from a recording of the wing stroke through the light sheet for each test. The Matlab program *PIVLab* was used to create a 52 × 52-point velocity vector field from image pairs. Results were analysed using direct Fourier transform correlation algorithm with a 64 × 64-pixel interrogation area with 50% overlap for cross-correlation. To achieve flow relative to the wing, the horizontal flow component induced by the wing's motion and the vertical flow component from the inflow perturbation depending on $J_{\text{Vert}}$ had to be added to the velocity vector field collected from the camera's static frame of reference. A total of 10 image pairs were collected per experiment which were phase averaged. Despite the low PIV sample size ($N = 10$), key flow features commonly noted on flapping wings such as LEVs were clearly discernible; however, further detailed measurements of the flow profile will be beneficial.

## 3. Results and discussion

### 3.1. Aerodynamic characteristics under a vertical inflow

#### 3.1.1. Mean lift and drag performance

In general, the mean lift and drag developed by the wing increased with increasing magnitude of upward inflow ($J_{\text{Vert}} > 0$) while the opposite trend was observed for downward inflows ($J_{\text{Vert}} < 0$) (figure 4*a*). This was expected due to the change in effective angle of attack of the wing, increasing for upflows and decreasing for downflows.

We compared the mean lift and drag during pronation, translation and supination phases of the stroke to identify potential effects of the inflow perturbations. For downward inflow perturbations, a reduction in the mean lift coefficient $\bar{C}_L$ occurred linearly with absolute inflow magnitude over the entire stroke including the rotation and translation phases (figure 4*b*). The mean drag $\bar{C}_D$ also exhibited linear variation during the translation phase of the stroke, whereas $\bar{C}_D$ was relatively constant during the supination phase for $-1 \leq J_{\text{Vert}} \leq 0$ and decreased rapidly during pronation as $J_{\text{Vert}}$ was varied from 0 to $-0.3$ (figure 4*c*).

For upward inflow perturbations, there was a deviation in the linear variation in the mean $\bar{C}_L$ and $\bar{C}_D$ during the translation and supination phases when $0.3 \leq J_{\text{Vert}} \leq 0.6$. The relative reduction in mean lift

and drag within this range of inflow ratios suggests that the wing experiences a reduction in effective angle of attack despite increase in inflow magnitude. Further increase in $J_{\text{Vert}}$ led to a significant variation in both $\bar{C}_L$ and $\bar{C}_D$ at $J_{\text{Vert}} = 0.7$ and recovery of the linear trend for $0.7 \leq J_{\text{Vert}} \leq 1$ (figure 4b,c). There was also a notable reduction in $\bar{C}_D$ during the pronation phase for $0.4 \leq J_{\text{Vert}} \leq 0.6$, roughly correlating with the reductions observed in the translation and supination phases.

### 3.1.2. Phase-averaged performance

To better understand the significance of the variations in mean lift and drag of the wing under different inflow conditions, the time-resolved forces were examined. The phase-averaged time traces of lift coefficient $C_L$ of the wing under quiescent conditions ($J_{\text{Vert}} = 0$) presented commonly known aerodynamic characteristics. These included a parabolic variation in the translation phase of the stroke accompanied with small peaks of lift created by rotational effects during the first rotation (pronation) and second rotation (supination) phases of each stroke (black line in figure 5a) which has been also observed by Dickinson et al. [26], Wu and Sun [27] and Bhat et al. [20]. The drag coefficient $C_D$ at $J_{\text{Vert}} = 0$ also displayed similar features to those of $C_L$ with larger peaks of drag during the first rotation phase of the stroke (black line in figure 5b) which is aligned with the measurements reported in Dickinson et al. [26].

For downward inflow perturbation conditions, the profile of the phase-averaged time traces of $C_L$ and $C_D$ matched those measured in quiescent conditions ($-1 \leq J_{\text{Vert}} < 0$) (figure 5a,b). The variation in $C_L$ and $C_D$ over the stroke was consistent for the downward inflows with proportional scaling down of forces when $|J_{\text{Vert}}|$ increased, which corresponded to the nominally linear reduction in both $\bar{C}_L$ and $\bar{C}_D$ (figure 4a). However, negative $J_{\text{Vert}}$ appeared to result in larger peaks of $C_L$ at the onset of supination phase and smaller peaks of $C_D$ during the pronation phase (figure 5a,b).

For upward inflow perturbations, based on the variation in the mean forces (figure 4), two different regimes were classified depending on the magnitude of the inflow ratio. In the first regime $R_1$, for $0 \leq J_{\text{Vert}} < 0.3$, the time-resolved $C_L$ displayed similar characteristics to those in quiescent conditions. In the second regime $R_2$ ($0.3 \leq J_{\text{Vert}} < 0.7$), the time history of $C_L$ changes in profile as shown in figure 5c. Here, the lift during the stroke is skewed with reduced lift during the first half of the stroke and a resurgence of lift during the second half, particularly at $J_{\text{Vert}} = 0.4$ (figure 5c). However, the variations of $C_L$ at $J_{\text{Vert}} = 0.5$ and 0.6 are more uniform, with increased lift production during the pronation phase of the stroke than $J_{\text{Vert}} = 0.4$ (figure 4b). Finally, for $0.7 \leq J_{\text{Vert}} \leq 1$, the profile of phase-averaged lift and drag once again resembles that measured during $R_1$ where the time-resolved $C_L$ has a symmetric and parabolic profile (figure 5c) and scales with the magnitude of the inflow. The variation of the drag coefficient $C_D$ was less dramatic compared with $C_L$ for the upwards inflow, but it was not as consistent as the variation in $C_D$ for the downward inflow (figure 5d).

## 3.2. Analysis of flow behaviour

### 3.2.1. Wake interaction

The flow profile around the wing obtained from DPIV is shown in figure 6, with the wing at the middle of the stroke. For $0 \leq J_{\text{Vert}} \leq 0.2$ where the first regime $R_1$ occurs, the flow velocity distribution ahead of the wing remains quiescent (figure 6a (i) and (ii)). However, when a moderate upwards inflow is induced ($0.3 \leq J_{\text{Vert}} \leq 0.6$ in the second regime $R_2$), the wake, which is normally shed beneath the wing, is translated upwards and into the path of the reciprocal stroke leading to the creation of a large downwards flow region around the wing (figure 6a (iii), (iv), (v) and (vi)). Within this range of inflow ratios, the flow ahead of the wing is dominated by downwards flow from the wake of the previous stroke which appears to adversely interact with the wing during subsequent strokes. The phase where the oncoming flow is dominated by the wake from the previous stroke also coincides with the skewed time-resolved $C_L$ and $C_D$ variation, (figure 5) as well as the trend deviations in mean $\bar{C}_L$ and $\bar{C}_D$ shown in figure 4. As the inflow ratio increased to $0.8 \leq J_{\text{Vert}} \leq 1$, the wake from the previous stroke appears to have been translated well above the wing's path during the subsequent stroke. Thus, flow upstream of the wing did not contain a significant downward component, suggesting that the wing was flapping through clean flow and undisturbed by the wake (figure 6a (vii) and (viii)). In this regime, the flow profile and interactions resemble $R_1$.

By contrast, for downwards inflow perturbations, there was no notable variation in the flow profile ahead of the wing's leading edge. Pronounced wing–wake interaction did not occur in this case as the inflow perturbation was oriented in the same direction of the wake, thus causing the wake to be

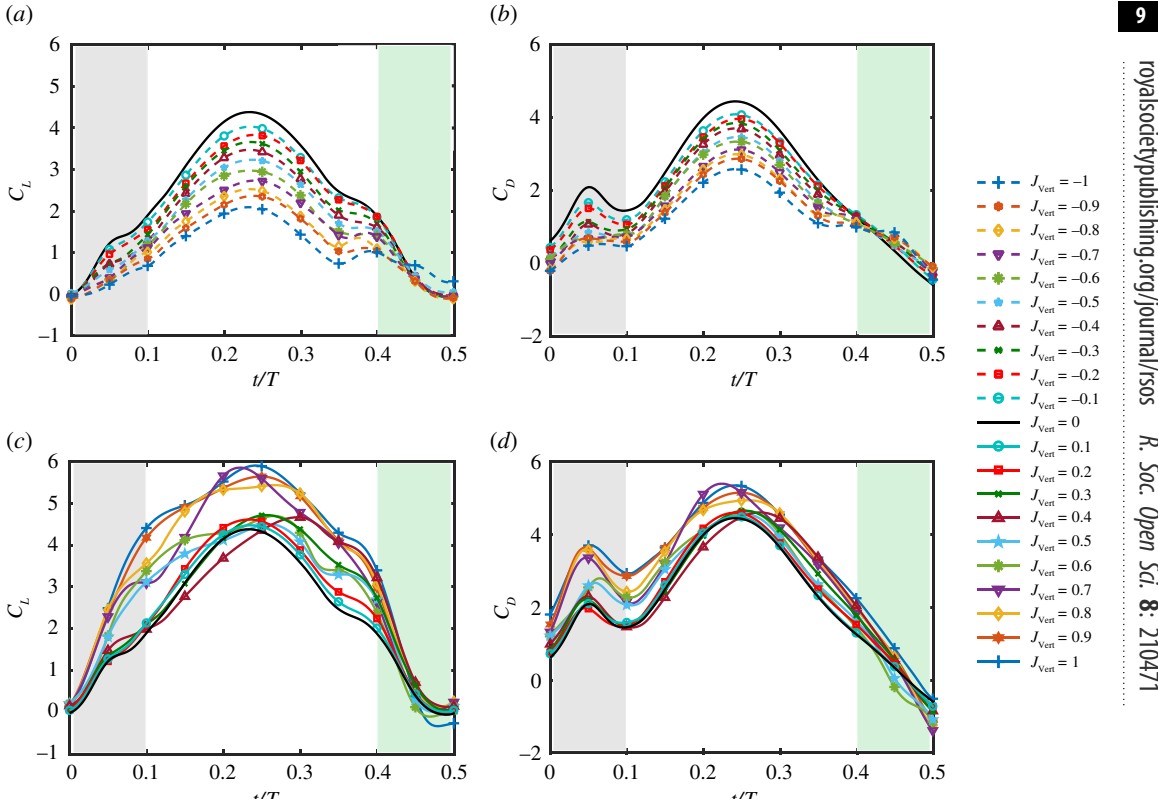

**Figure 5.** The phase-averaged time traces of $C_L$ and $C_D$. Lift and drag coefficients $C_L$ and $C_D$ for (a) and (b) downward inflow perturbations. (c) and (d) upward inflow perturbations. Grey, white and green backgrounds mark pronation, translation and supination phases of the stroke, respectively.

advected further downstream from the wing (figure 6b). The consistency of the interaction trend between the inflow ratios in the absence of any pronounced wake effects resulted in the nominally linear variation in lift and drag as downward flow velocity was increased (figure 4b,c). Additionally, Dickinson *et al.* [26] and Wu and Sun [27] suggested the wake produced during the stroke reversal can cause momentary increase in the lift and drag due to circulation effects. However, the circulation effects suggested by the previous studies were diminished for downwards inflow because the wake created during stroke reversal was probably translated well below the stroke path. As a result, the lift and drag during the first rotation phase of the stroke experienced a reduction (figure 4c and figure 5b).

### 3.2.2. Effective angle of attack

The effect of the wake interaction is further explored here in terms of effective angle of attack ($\Theta_{\text{eff}}$), which is the angle of the incoming flow relative to the wing's pitch angle in the translation phase of the stroke ($\Theta_{\text{max}}$). The $\Theta_{\text{eff}}$ was determined from the velocity vector arrays obtained from DPIV images as shown in equation (3.1).

$$\Theta_{\text{eff}} = \Theta_{\text{max}} + \tan^{-1}(J_{\text{Vert}_{\text{eff}}}), \tag{3.1}$$

where $J_{\text{Vert}_{\text{eff}}} = \tan^{-1}(v/w)$ is the average angle of the flow in the wing reference frame along the right-hand edge of the DPIV image window located 0.3 chord lengths ahead of the leading edge of the wing (figure 6a (iv)). The theoretical angle of attack ($\Theta_{\text{th}}$) was calculated based on the known velocity of the wing and inflow using equation (3.2).

$$\Theta_{\text{th}} = \Theta_{\text{max}} + \tan^{-1}(J_{\text{Vert}}). \tag{3.2}$$

For downwards inflows, due to the absence of wake interaction, the effective angle of attack ($\Theta_{\text{eff}}$) matched the theoretical angle of attack ($\Theta_{\text{th}}$) (figure 7). It can be seen that $\Theta_{\text{eff}}$ decreases linearly as $J_{\text{Vert}}$ becomes more negative, thereby creating a linear trend in $\bar{C}_L$ and $\bar{C}_D$ (figure 4b,c).

Despite $J_{\text{Vert}} = -1$ having $\Theta_{\text{eff}} \approx 0$ at the $R_{\text{RoG}}$, the wing still produces lift over its stroke under the downwards inflow (figure 4b), presumably from the increased velocity at the outboard section of the

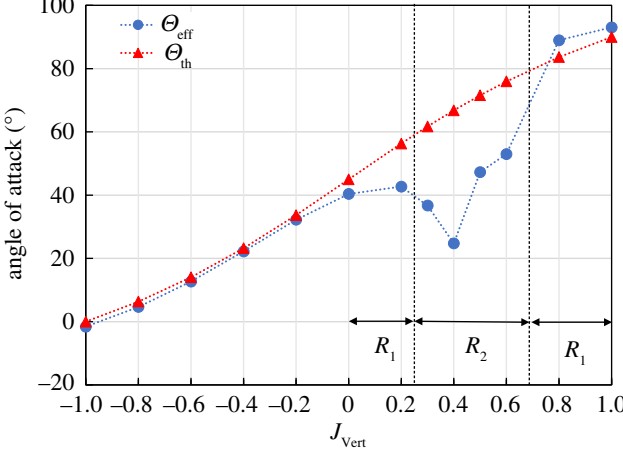

(a) (i)    (ii)    (iii)    (iv)

0.3c

$J_{Vert} = 0$    $J_{Vert} = 0.2$    $J_{Vert} = 0.3$    $J_{Vert} = 0.4$

(v)    (vi)    (vii)    (viii)

$J_{Vert} = 0.5$    $J_{Vert} = 0.6$    $J_{Vert} = 0.8$    $J_{Vert} = 1$

normalized vertical velocity ($v/U_{RoG}$)

0    −0.5    −1.0    −1.5    −2.0

(b) (i)    (ii)    (iii)    (iv)

$J_{Vert} = -0.2$    $J_{Vert} = -0.4$    $J_{Vert} = -0.8$    $J_{Vert} = -1$

**Figure 6.** Vertical component of the flow phase averaged over 10 DPIV image pairs. The vertical velocity colour map at $t/T = 0.25$ normalized with $U_{RoG}$ under (a) upward inflow for i, ii, vii and viii in the regime $R_1$ and iii–vi in the regime $R_2$. (b) downward inflow. The vertical velocity colour map captures wake by highlighting the downwards flow with the velocity vectors displaying flow velocity from a static frame of reference. Due to negligible upwards flow the colour bar is one-sided and limits the contour map to downwards flow.

**Figure 7.** Effective and theoretical angle of attack. The actual $\Theta_{eff}$ was found using the average angle of the incoming flow on the right edge of the DPIV domain, which is $0.3c$ distance in front of the leading edge of the wing (see inset in figure 6a(iv)).

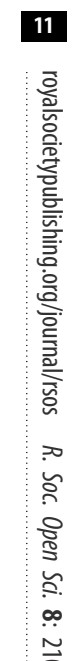

**Figure 8.** Vortex identification. Phase-averaged Q-criterion at $t/T = 0.25$ normalized with $R_{RoG}$ and $U_{RoG}$ under ($a$) upward inflow for i, ii, vii and viii in the regime $R_1$ and iii–vi in the regime $R_2$. ($b$) downward inflow.

wing allowing the wing to maintain a positive angle of attack in that portion of the wing. Potentially, if the inflow magnitude is equal to the velocity at the wing's tip, this will cause a negative $\Theta_{eff}$ across the entirety of the wing causing a complete loss of lift. This suggests that in order to maintain lift during such downwards inflow perturbations, the wing must maintain a positive angle of attack on the flow either by increasing its flapping frequency to increase the velocity of the wing, or, increase the wing's pitch angle as is done in an ascending manoeuvre [28].

Considering the effect of upward-oriented inflow perturbations ($J_{Vert} > 0$ in figure 7), there is a clear reduction in the measured effective angle of attack $\Theta_{eff}$ compared with the theoretical angle of attack $\Theta_{th}$ from $J_{Vert} = 0$ up to $J_{Vert} = 0.8$. The reduction in $\Theta_{eff}$ for $J_{Vert} = 0$ condition may be attributed to the wake from the previous stroke. As the magnitude of upwards inflow increases ($J_{Vert}$ becoming more positive), the effects of the wake dominates flow around the wing as highlighted in figure 6$a$ (ii) and (iii), leading to a further decrease in $\Theta_{eff}$, with the minimum $\Theta_{eff} = 24.7°$ at $J_{Vert} = 0.4$ (figure 7). The increase in the effective angle of attack $\Theta_{eff}$ at $J_{Vert} = 0.8$ ($\Theta_{eff} = 88.9°$) to nominally the theoretical angle of attack ($\theta_{th} = 83.7°$) demonstrates that the wake from the previous stroke has been translated by the inflow and no significant wake interaction occurs as shown previously in figure 6$a$ (vii). Thus, when the wing flaps in the first regime $R_1$ under the upward inflow, $\Theta_{eff}$ shows a reasonable agreement with $\Theta_{th}$, whereas there is a significant deviation from $\Theta_{th}$ in the second regime $R_2$.

### 3.2.3. Flow profile over the wing

The vorticity field was statistically characterized to demonstrate the existence of LEV using Q-criterion method [29,30]. A vortex was identified based on areas where the magnitude of the rotation rate tensor ($[\Omega] = 1/2(\nabla \mathbf{v} - (\nabla \mathbf{v})^T)$) was greater than the magnitude of the strain rate tensor ($[\psi] = 1/2(\nabla \mathbf{v} + (\nabla \mathbf{v})^T)$) which led to a positive value of $Q$ calculated as ($1/2[\|[\Omega]\|^2 - \|[\psi]\|^2]$), where

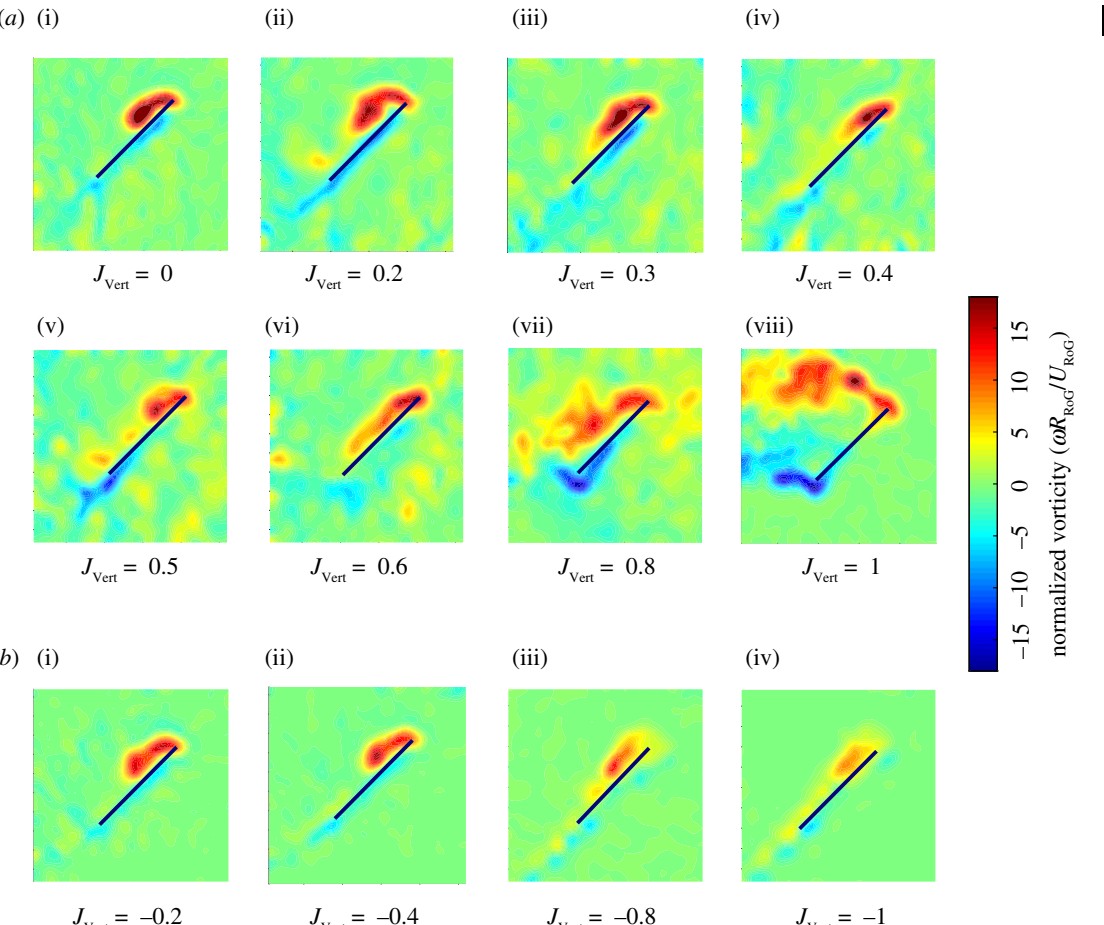

**Figure 9.** Phase-averaged vorticity field based on 10 DPIV image pairs at mid-stroke. The non-dimensional vorticity field at middle of the stroke extracted from DPIV under (*a*) upward inflow for i, ii, vii and viii in the regime $R_1$ and iii–vi in the regime $R_2$. (*b*) downward inflow.

the double vertical bars denote Frobenius norm. The antisymmetric part ([$\Omega$]) and symmetric part ([$\psi$]) of the velocity gradient tensor were calculated for every point within each $52 \times 52$-point velocity vector field from DPIV image pairs. Figure 8 exhibits $Q > 0$ in all inflow ratios which indicates the presence of LEV.

The overall chordwise vorticity distribution at middle of the stroke in the quiescent case ($J_{\mathrm{Vert}} = 0$; figure 9*a* (i)) shows characteristics of the presence of a LEV, which is a stable vortex that spirals outwards along the leading edge of the wing keeping flow attached to the wing [31,32]. A number of previous studies have linked the strength of the LEV to the effective angle of attack [9,13] and the variations in the effective angle of attack $\Theta_{\mathrm{eff}}$ appears to match with the LEV size here. The reduction in the effective angle of attack $\Theta_{\mathrm{eff}}$ due to the effects of wake in regime $R_2$ under the upward inflow appears to result in smaller LEV forming over the wing (figure 9*a* (iii), (iv), (v) and (vi)), which could explain the reduction in mean lift noted in this regime (figure 4). At $J_{\mathrm{Vert}} = 1$, vortices are being shed behind the wing at both its leading edge and trailing edge due to the $\Theta_{\mathrm{eff}}$ of approximately 90° (at $R_{\mathrm{RoG}}$) (figure 9*a* (viii)) allowing for substantial amounts of lift to be produced across the entire stroke (figure 5) as the wake is no longer present to decrease $\Theta_{\mathrm{eff}}$. It should be noted that this lift is essentially bluff body drag and comes with a large drag penalty [33].

The LEV appears to diminish in magnitude continuously as the downward inflow ($|J_{\mathrm{Vert}}|$) increased due to the continuous reduction in the effective angle of attack $\Theta_{\mathrm{eff}}$ (figure 9*b*). The reduction in the size of the LEV led to overall reduction in lift and drag by the wing under the downward inflow (figure 4*a*).

### 3.2.4. Variation in lift production per stroke

To better understand the transient effects of the wing–wake interaction on the lift produced by the flapping wing, we compare the time-resolved $C_L$ per stroke obtained from the different tests used for phase averaging to assess the variation in $C_L$ between individual strokes for each $J_{\mathrm{Vert}}$ (figure 10).

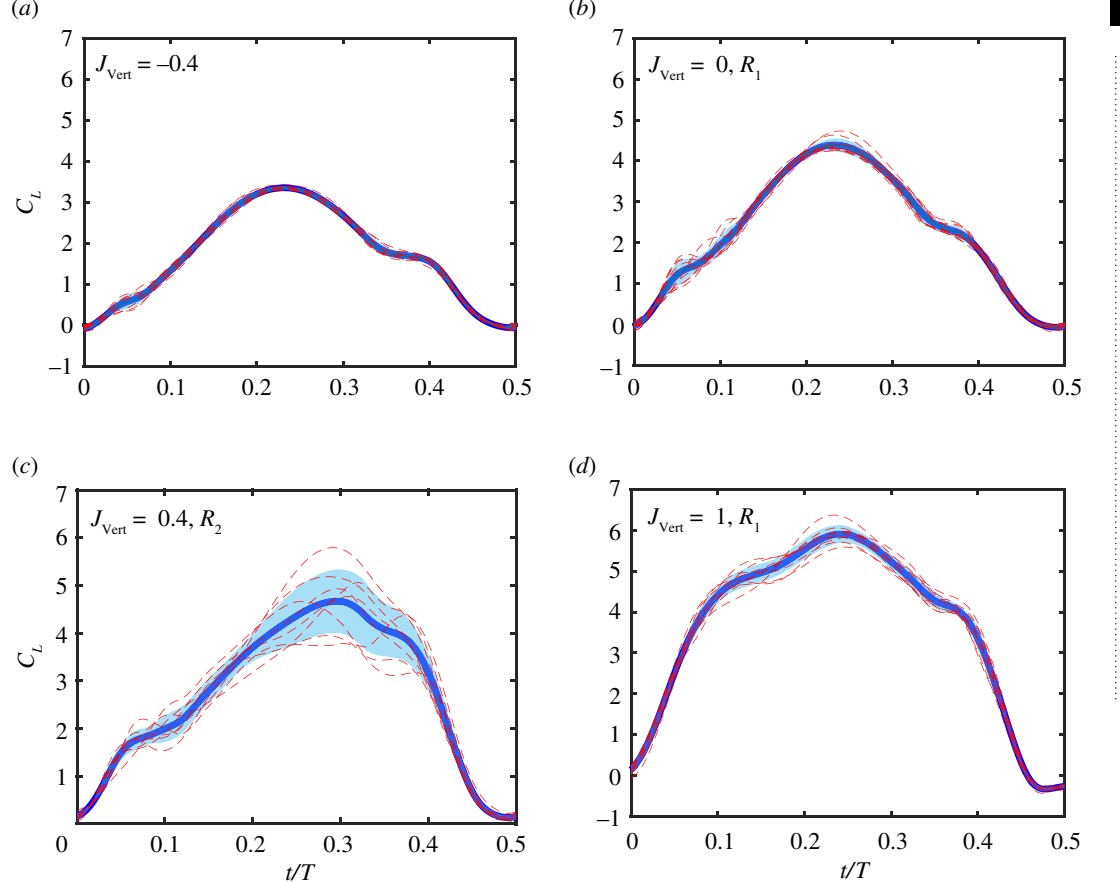

**Figure 10.** Stroke-by-stroke lift variation. Phase-averaged $C_L$, time-resolved standard deviation $\sigma$, and individual stroke $C_L$ curves for (a) $J_{\text{Vert}} = -0.4$. (b) $J_{\text{Vert}} = 0$. (c) $J_{\text{Vert}} = 0.4$. (d) $J_{\text{Vert}} = 1$. Thick lines denote phase-averaged $C_L$ while dashed red lines are $C_L$ curves from individual tests. The shaded area denotes one standard deviation.

We also compare the time-resolved standard deviations for the stroke given by $\sigma = \sqrt{\sum_{i=1}^{N} (C_L(i) - \bar{C}_L)^2 / N}$, where $N$ is number of the tests, for different inflow ratios.

For all downward inflow perturbations as well as upward inflows in the first regime, stroke lift coefficients are consistent across the different flapping stroke with a small standard deviation $\sigma$ (figure 10$a$,$b$ and $d$). By contrast, in the second regime, $R_2$ for upward inflow conditions, inconsistent lift variations per stroke across the tests with a high standard deviation was noted, figure 10$c$. This variation is probably due to the interaction between the wing and the turbulence in the wake from the previous stroke and highlights the transient effects of interactions with the wake. Further details on the variation in $C_L$ and $C_D$ between strokes for different inflow ratios can be found in the electronic supplementary material, figure S1.

# 4. Conclusion

The performance of a flapping wing under a uniform vertical inflow perturbation was studied experimentally using a Reynolds-scaled flapping wing apparatus performing prescribed flapping motion with pitching angle 45° and flapping amplitude of 120° at Reynolds number of 3600. The overall lift and drag forces were measured by a F/T sensor and the flow profile over the wing was captured using two-dimensional DPIV, while the magnitude and orientation of the vertical inflow were varied by the inflow ratio, the ratio of the induced inflow velocity to the wing velocity for $-1 \leq J_{\text{Vert}} \leq 1$.

For downward inflows, the mean lift and drag coefficients ($\bar{C}_L$ and $\bar{C}_D$) were found to decrease linearly with increasing inflow ratio ($|J_{\text{Vert}}|$).

Compared with quiescent conditions ($J_{\text{Vert}} = 0$), the mean lift and drag coefficients ($\bar{C}_L$ and $\bar{C}_D$) tended to increase as the magnitude of the upward inflow increased. However, within the range of $0.3 \leq J_{\text{Vert}} < 0.7$, $\bar{C}_L$ and $\bar{C}_D$ deviated from the linear trend with lower proportional increase with the increase in inflow

magnitude. Analysis of the flow profile around the wing revealed that in this range of upward inflows, the wake of the wing was translated upwards, due to the inflow perturbation and into the path of the reciprocal stroke. This led to a notable downward-oriented flow around the wing's leading edge which resulted in adverse interaction and diminished lift and drag. The effective angle of attack ($\Theta_{\text{eff}}$, determined from the velocity vector arrays obtained from DPIV images) was found to be strongly affected by the downwards flow from the wake of the previous stroke, such that it significantly deviated from the theoretical angle of attack ($\Theta_{\text{th}}$, calculated based on the known velocity of the wing and inflow) resulting in significant variations in the stroke-to-stroke variations in the lift and drag coefficients when $0.3 \leq J_{\text{Vert}} < 0.7$.

For $J_{\text{Vert}} \geq 0.7$, the wake from the previous stroke was translated well above the wing's path during the subsequent stroke. Thus, the flow upstream of the wing did not contain a significant downward component and caused an increase in $\Theta_{\text{eff}}$, with $\bar{C}_L$ and $\bar{C}_D$ assuming the nominally linear trend once again.

Data accessibility. Dataset and source code for this work are available at the Dryad Digital Repository: https://doi.org/10.5061/dryad.1g1jwstvz [34].

Authors' contributions. The experiments were performed by J.S. and S.M. The experiments were conceptualized and designed by J.S., S.M., A.M. and S.R. All authors contributed to the interpretation of findings, drafting and revising the manuscript.

Competing interests. We declare we have no competing interests.

Funding. This work was supported by Air Force Office of Scientific Research grants, FA2386-20-1-4084, FA2386-19-1-4066 and Australian Research Council Discovery Project (project no DP200101500).

Acknowledgements. We wish to thank the Technical Support Group (TSG) at UNSW Canberra for their assistance with equipment and useful advice for fabrication of the flapping rig. The authors thank Mark Shortis for his assistance with using high-speed camera during the experiments.

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
