## [Peer Review File · Royal Society Open Science]

Review History

RSOS-210471.R0 (Original submission)

Review form: Reviewer 1

Is the manuscript scientifically sound in its present form?

Yes

Are the interpretations and conclusions justified by the results?

Yes

Is the language acceptable?

Yes

Do you have any ethical concerns with this paper?

No

Have you any concerns about statistical analyses in this paper?

No

Recommendation?

Accept with minor revision (please list in comments)

Comments to the Author(s)

This revised manuscript is currently under consideration for Open Science after being transferred from Interface. The authors answered most of my questions. Here are some concerns to be addressed based on the answers the authors provided to my questions. The numbers corresponds to the original question number in the 1st round of revision (see Appendix A).

2. I am unclear about how the authors define gust. On one hand, they are motivated by gusty winds but on the other hand, they reply that "majority of the perturbations encountered in realistic environments are likely to be of timescales much lower than those of the wingbeat..." therefore, they assume "wing flapping frequency and simulated as a uniform constant velocity perturbation in the freestream flow". Therefore, in this research gust is not a factor? Why mention it or discuss it? Why not simply remove the entire discussion on gust as it is irrelevant to the studied topic.

4. PIV: How do you decide what is "significant variation between image pairs for each phase were not noted"? Was this a visualize inspection or a statistical test. 10 image pairs seems to be a very low number to enable capturing any dynamic phenomena associated with fluid mechanics, specifically the formation of LEV .

PIV setup: 4 mm light sheet seems a bit large that can cause significant out of plane motion for the particles while moving within the light sheet, was this issue recognized?

6. Wing-wake interaction: It is still unclear how the authors based on visualize inspection of the velocity field determines whether interaction occurs or not. Interaction can occur within the velocity gradients; for example between the vorticity and stress fields (see Tsinober A, 2001, Introduction to Turbulence). A statistical analysis that provide some insight into the skewness of the interaction between the velocity gradients may indicate the existence of such interaction is only one option out of many procedures that provide a quantitative observation.

Review form: Reviewer 2

Is the manuscript scientifically sound in its present form?

Yes

Are the interpretations and conclusions justified by the results?

Yes

Is the language acceptable?

No

Do you have any ethical concerns with this paper?

No

Have you any concerns about statistical analyses in this paper?

No

Recommendation?

Major revision is needed (please make suggestions in comments)

Comments to the Author(s)

Please see the attachment (Appendix B).

Decision letter (RSOS-210471.R0)

Dear Dr Mazharmanesh

The Editors assigned to your paper RSOS-210471 "Effects of Uniform Vertical Inflow Perturbations on the Performance of Flapping Wings" have now received comments from reviewers and would like you to revise the paper in accordance with the reviewer comments and any comments from the Editors. Please note this decision does not guarantee eventual acceptance.

Please submit your revised manuscript and required files (see below) no later than 21 days from today's (ie 19-Apr-2021) date. Note: the ScholarOne system will 'lock' if submission of the revision is attempted 21 or more days after the deadline. If you do not think you will be able to meet this deadline please contact the editorial office immediately.

on behalf of Professor Kevin Padian (Subject Editor)
openscience@royalsociety.org

Associate Editor Comments to Author:

Thank you for the close attention you have paid to the comments of the reviewers at JRSI. A few matters remain to be addressed, including a further linguistic polish - it may be worthwhile taking advantage of one of the services at <https://royalsociety.org/journals/authors/benefits/language-editing/> for this support. We'll look forward to receiving a revision in due course.

Reviewer comments to Author:

Reviewer: 1

Comments to the Author(s)

This revised manuscript is currently under consideration for Open Science after being transferred from Interface. The authors answered most of my questions. Here are some concerns to be addressed based on the answers the authors provided to my questions. The numbers corresponds to the original question number in the 1st round of revision.

2. I am unclear about how the authors define gust. On one hand, they are motivated by gusty winds but on the other hand, they reply that "majority of the perturbations encountered in realistic environments are likely to be of timescales much lower than those of the wingbeat..." therefore, they assume "wing flapping frequency and simulated as a uniform constant velocity perturbation in the freestream flow". Therefore, in this research gust is not a factor? Why mention it or discuss it? Why not simply remove the entire discussion on gust as it is irrelevant to the studied topic.

4. PIV: How do you decide what is "significant variation between image pairs for each phase were not noted"? Was this a visualize inspection or a statistical test. 10 image pairs seems to be a very low number to enable capturing any dynamic phenomena associated with fluid mechanics, specifically the formation of LEV .

PIV setup: 4 mm light sheet seems a bit large that can cause significant out of plane motion for the particles while moving within the light sheet, was this issue recognized?

6. Wing-wake interaction: It is still unclear how the authors based on visualize inspection of the velocity field determines whether interaction occurs or not. Interaction can occur within the velocity gradients; for example between the vorticity and stress fields (see Tsinober A, 2001, Introduction to Turbulence). A statistical analysis that provide some insight into the skewness of the interaction between the velocity gradients may indicate the existence of such interaction is only one option out of many procedures that provide a quantitative observation.

Reviewer: 2

Comments to the Author(s)

Please see the attachment.

===PREPARING YOUR MANUSCRIPT===

===PREPARING YOUR REVISION IN SCHOLARONE===

- Any electronic supplementary material (ESM).
- If you are requesting a discretionary waiver for the article processing charge, the waiver form must be included at this step.
- If you are providing image files for potential cover images, please upload these at this step, and inform the editorial office you have done so. You must hold the copyright to any image provided.
- A copy of your point-by-point response to referees and Editors. This will expedite the preparation of your proof.

- Ensure that your data access statement meets the requirements at <https://royalsociety.org/journals/authors/author-guidelines/#data>. You should ensure that you cite the dataset in your reference list. If you have deposited data etc in the Dryad repository, please include both the 'For publication' link and 'For review' link at this stage.
- If you are requesting an article processing charge waiver, you must select the relevant waiver option (if requesting a discretionary waiver, the form should have been uploaded at Step 3 'File upload' above).
- If you have uploaded ESM files, please ensure you follow the guidance at <https://royalsociety.org/journals/authors/author-guidelines/#supplementary-material> to include a suitable title and informative caption. An example of appropriate titling and captioning may be found at https://figshare.com/articles/Table_S2_from_Is_there_a_trade-off_between_peak_performance_and_performance_breadth_across_temperatures_for_aerobic_scooping_in_teleost_fishes_/3843624.

Author's Response to Decision Letter for (RSOS-210471.R0)

See Appendix C.

RSOS-210471.R1 (Revision)

Review form: Reviewer 1

Is the manuscript scientifically sound in its present form?

Yes

Are the interpretations and conclusions justified by the results?

Yes

Is the language acceptable?

Yes

Do you have any ethical concerns with this paper?

No

Have you any concerns about statistical analyses in this paper?

Yes

Recommendation?

Accept as is

Comments to the Author(s)

My concerns has been answered partially because 10 image pairs are not sufficient to deduce any meaningful insight on the fluid dynamics of the phenomena. Yet, I do understand the complexity of performing new experiments. Perhaps, a clear statement by the authors about the validity of their observation that is sensitive to the lack of statistical convergence.

Review form: Reviewer 2

Is the manuscript scientifically sound in its present form?

Yes

Are the interpretations and conclusions justified by the results?

Yes

Is the language acceptable?

Yes

Do you have any ethical concerns with this paper?

No

Have you any concerns about statistical analyses in this paper?

Yes

Recommendation?

Accept with minor revision (please list in comments)

Comments to the Author(s)

Please see my attachment (Appendix D).

Decision letter (RSOS-210471.R1)

Dear Dr Mazharmanesh

On behalf of the Editors, we are pleased to inform you that your Manuscript RSOS-210471.R1 "Effects of Uniform Vertical Inflow Perturbations on the Performance of Flapping Wings" has been accepted for publication in Royal Society Open Science subject to minor revision in

accordance with the referees' reports. Please find the referees' comments along with any feedback from the Editors below my signature.

Please submit your revised manuscript and required files (see below) no later than 7 days from today's (ie 26-May-2021) date. Note: the ScholarOne system will 'lock' if submission of the revision is attempted 7 or more days after the deadline. If you do not think you will be able to meet this deadline please contact the editorial office immediately.

on behalf of Prof Kevin Padian (Subject Editor)
openscience@royalsociety.org

Associate Editor Comments to Author:

The reviewers have a few remaining comments notably ensuring some qualification is made regarding the sample size of the study. While the journal will conduct a 'light touch' copyediting during typesetting, you may benefit from seeking additional professional language editing guidance (as has been noted previously).

Editor comments:

Thank you for your revisions. Please address the remaining concerns of our reviewer when you submit your final version. Best wishes.

Reviewer comments to Author:

Reviewer: 1

Comments to the Author(s)

My concerns has been answered partially because 10 image pairs are not sufficient to deduce any meaningful insight on the fluid dynamics of the phenomena. Yet, I do understand the complexity of performing new experiments. Perhaps, a clear statement by the authors about the validity of their observation that is sensitive to the lack of statistical convergence.

Reviewer: 2

Comments to the Author(s)

Please see my attachment.

===PREPARING YOUR MANUSCRIPT===

===PREPARING YOUR REVISION IN SCHOLARONE===

-- If you have uploaded ESM files, please ensure you follow the guidance at <https://royalsociety.org/journals/authors/author-guidelines/#supplementary-material> to include a suitable title and informative caption. An example of appropriate titling and captioning may be found at https://figshare.com/articles/Table_S2_from_Is_there_a_trade-off_between_peak_performance_and_performance_breadth_across_temperatures_for_aerobic_scops_in_teleost_fishes_/3843624.

Author's Response to Decision Letter for (RSOS-210471.R1)

See Appendix E.

Decision letter (RSOS-210471.R2)

Dear Dr Mazharmanesh,

I am pleased to inform you that your manuscript entitled "Effects of Uniform Vertical Inflow Perturbations on the Performance of Flapping Wings" is now accepted for publication in Royal Society Open Science.

Follow Royal Society Publishing on Twitter: @RSocPublishing
Follow Royal Society Publishing on Facebook:
<https://www.facebook.com/RoyalSocietyPublishing>
Read Royal Society Publishing's blog:
<https://royalsociety.org/blog/blogsearchpage/?category=Publishing>

Appendix A

The paper studies the effects of vertically oriented perturbations on the performance of flapping wings inspired by insects' flight in a windy environment. The authors performed controlled study on an airfoil wing mounted on servo motor mimicking heaving and pitching motion as well as vertical motion of the wing which supposed to mimic upward and downward wind perturbation (i.e.: gusts). Force measurements are taken through force transducer and PIV was employed to describe the wake flow developed by these motions.

I have few conceptual reservations:

1. The experiments are performed in quiescent flow; thus, no mean shear (i.e.: wind). If there is no mean shear, how does this experiment resemble to the so-called In-vivo conditions or to real flow scenarios?
2. The authors are motivated by inflow perturbations (gusts), abruptly occurring in windy environment. These abrupt gusts are unpredictable and their magnitude sometimes may be very small and sometime large in comparison to the mean flow. The experiments here do not simulate any type of perturbations. The experiments include moving the wing up and down at different rates; these can be associate with ascending or descending of the object (either an organism or a UAV) during flight; meaning this is more similar to gaining altitude or losing altitude during flight; the connection to perturbations/gusts/fluctuations is far-stretched if none.
3. Finally, the PIV experiments potentially can provide a wider base of analysis. The authors chose to show few instantaneous/mean velocity and vorticity maps as function of J_{vert} without any rigorous statistical attempt to identify LEV, characterize wing-wake interaction (i.e.: velocity gradients, velocity profiles or kinetic energy) and seems very loose mainly in comparison to the force balance measurements.

Some specific comments:

1. The Reported Reynolds number is 3600 suggested by the authors to corresponds to “*large insects*”. The Reynolds number follow Weisfogh (1973) definition. Yet, Weisfogh suggest that this Reynolds number corresponds to “*tiny insects*” (see page 173, before equation 10 in his paper). Since the authors do not specify the insect in study and keep it generic: bees, wasp...and there is range of sizes amongst these species, this discrepancy should be revised.
2. “*Relating this to flapping wing insects for bees and wasps which operate at approximately $Re=3600$, $J_v = \pm 1$ is equivalent to a wind speed of approximately 5.2 m/s, comparable to the maximum operating wind speed of 6 – 8 m/s for bees and wasps..*”
3. This statement is somewhat misleading. The authors invert the Reynolds number based on the rotational speed to suggest it is equivalent to wind speed. Yet, during the rotational motion of the wing, the tangential velocity experience instantaneous acceleration due to the change in rotational motion. This requires at least some explanation in the text.
4. PIV:
How many PIV images were acquired per test run?
What was the camera resolution used in the PIV experiments?
Many details of the PIV setup are missing.

5. Are the PIV velocity maps instantaneous in figure 6?

Figure 6 depict a) as upward inflow and b) as downward inflow. It is somewhat confusing: is the upward means that the wing moves up? If so, why most of the region above and below the wake show negative values of vertical velocity (blue color)? If the motion is downward, then it makes sense that fluid is carried downward. However, for $J_{\text{vert}} = -0.2$, it's a bit unclear why there is negative region. It could be that the normalization with the wing rotational speed change the signs?

6. Page 14, line 59: "Wing-wake interaction did not occur..." How can the authors decide based on the velocity field (instantaneous, I presume) about wing-wake interaction? Just because the flow is not skewed, you can speculate about the interaction?

7. Page 15, line 5: Figure 6 and not 7?

8. Page 15, line 5: "*The consistent interaction between the wing and inflow perturbation in the absence of any wake effects results in the nominally linear variation in lift and drag as downward flow velocity was increased (Figure 5 (b) and (c)).*"

9. What is consistent interaction stands for? What does it mean and how did the authors decided based on figure 6, which essentially only show one velocity component, that the interaction is consistent? Furthermore, does the vertical velocity marks wake effects? If the authors would have calculated the velocity gradients (i.e.: vorticity) and perhaps show rms or similar, then some qualitative perception of wake effects could be demonstrated. Where do the authors observe linear variation in figure 5c? Seems more like polynomial profile and less linear.

10. I also find the next sentence speculative when suggesting "*momentary increase in lift and drag over flapping wings*", where the author point to figure 6B that essentially does not show much and figure 5c that shows how lift vary over time due to the coupled motion interaction between rotating wing and moving it vertically. The so-called momentary increase is not quantified and can be associated with the system inaccuracies or with the kinematic motion that is coupled and generates such trend.

11. Page 17, line 10: It is unclear how was the effective angle of attack was determined from the PIV images? Based on how many images? Which velocity component was considered? If the vertical velocity was considered, what was the role of the horizontal velocity?

12. Page 19, line 8: "*...showed well-known LEV formation, a stable vortex that spirals outwards along the leading edge of the wing keeping flow attached to the wing..*"

13. Why do the authors decide that an instantaneous or mean vorticity map represents an LEV? LEV is a coherent pattern in the flow that plays a role in augmenting lift (Ben-Gida et al., 2020). In order to show that a concentrated region of vorticity that developed over a wing during flapping is an LEV, one need to statistically characterize the vorticity field to show its existence rather than a concentrated region of vorticity. For example, using some vortex identification

method and so on (Lambert et al., 2019). Without performing some sort of identification procedure, I find section 3.2.3 questionable.

14. The last section in the conclusion does not provide any meaningful information

References:

Ben-Gida, H., Gurka, R. and Weihs, D., 2020. Leading-edge vortex as a high-lift mechanism for large-aspect-ratio wings. *AIAA Journal*, 58(7), pp.2806-2819.

Lambert, W.B., Stanek, M.J., Gurka, R. and Hackett, E.E., 2019. Leading-edge vortices over swept-back wings with varying sweep geometries. *Royal Society open science*, 6(7), p.190514.

Appendix B

Review of manuscript for R. Soc. Open Sci.

Effects of Uniform Vertical Inflow Perturbations on the Performance of Flapping Wings

by Mazharmanesh *et al.*

I wish to thank the authors for responding to my previous comments. Please find included my review of the new manuscript for *Royal Society Open Science*.

General remarks

Positives:

- The manuscript is better overall.
- It seems appropriate for *Royal Society Open Science*.

Issues:

- There are some unresolved technical points.
- The writing needs to be more precise overall.
- Certain figures need more work (mainly formatting).

In summary: the manuscript has improved but numerous issues remain. I can only recommend publication once these have been addressed fully and satisfactorily.

Please accept advanced apologies for any errors, misinterpretations or omissions on my part.

General points

- Please check all writing again very carefully for precision and clarity.
- Please check all figures for clarity, legibility and consistency.
- Please bear in mind how the manuscript will appear when printed in hardcopy, particularly for those with diminished eyesight.
- There are instances of “et al.” (at least in the **Introduction**) that still need to be italicised.
- Small point: bracketed manufacturer details often include the company location (city and country). Please double-check this with the Editors.

Specific points

Introduction

- The new opening sentences position the study better. But I suggest rewording the first sentence such that “their” refers grammatically to “insects” and not “agility and manoeuvrability”, even if the intended meaning is clear from the context.
- **[line 64 onwards]** The new wording implies that there exist other categories of study beyond *numerical* and *experimental*. Please rephrase accordingly.
- **[line 80]** “...in two-wing miniature flying vehicles.” This seems somewhat incongruous. I suggest moving it up or just deleting it.

Experimental set-up and procedure

- **[Figure 1]** This figure is clearer than before.
 - o While I appreciate the left-hand image probably originated in CAD and a full vector rendition may not be possible, please ensure the final resolution of the raster elements is high.
 - o Please consider homogenising the appearance of the text and arrows across the figure, perhaps converting them all to vector format. Dimension units may not be needed on the left-hand side if the caption has them.
- **[line 167]** “The wing was...”. This sentence could still explain that the force coefficients level off after $h/c \geq 1.8$, even if **response-Figure 1** is omitted from the manuscript. Perhaps that figure could go in the Supplementary Material (SM) instead.
- **[Figure 2]**
 - o Presumably “RAD” (y-axis) denotes *radians*? May I suggest *degrees* instead? Indeed, both ϕ and θ have units of *degrees* in the nomenclature.
 - o The GoPro data are instructive; the stroke motion profile in **response-Figure 3** looks fairly sinusoidal. Do these data pertain to zero inflow? Might vertical translation of the flapper affect the stroke and/or pitch kinematics?
- **[line 198]** “A J_{vert} of up to and including 1...” again sounds a bit clunky. This sentence might be deleted altogether.
- **[Figure 3]**
 - o If this cannot be made vector, please increase the resolution.
 - o The text labels are a bit small.
- **[line 246]** “...phase-averaged forces...”. Apologies if I missed this, but how many strokes were used to phase-average the forces? Presumably 10, as for PIV? If missing, please add this information somewhere.

- **[line 266 onwards]** “Results were analysed...”. Perhaps this should move up the paragraph for better ordering.
- **[line 169]** “This sample size was deemed sufficient as significant variation between image pairs for each phase were not noted.”. This implies that the vector fields were inspected visually, which alone would be insufficient justification for deciding the number of PIV vector fields. But there is perhaps a more fundamental issue: the results figures (notably **Figure 9**) later show that the time-resolved force profile varies between strokes, particularly for inflows in the R_2 regime. This means the flow field varies as well—often in turbulent fashion (**line 464**). Are just 10 PIV realisations enough to provide robust statistical estimates of the mean flow in turbulent conditions?

Results and discussion

- **[line 286]** “...decreased rapidly during pronation...”. This seems like the same issue as before (concerning $-J_{vert}$). Please check this and other instances carefully again.
- **[line 311]** “...instantaneous...”. Not phase-averaged?
- **[line 330]** Please change “however” to “but”.
- **[Figure 5]** This figure is now easier to read, but please address the caption—it mentions both *phase-averaged* and *instantaneous*, which strictly are different.
- **[line 345]** “...appears to be translated upwards...”. This flow feature is at the crux of the study. Is it not possible to be more definitive?
- **[line 346]** “...downward flow region...”. With my previous comment about reference frames, I was referring to the reference frame of the *vector fields* in **Figure 6**, not the panels in the figure. Nonetheless, the addition of roman numerals is helpful.
- **[Figure 6]** The vector fields are clearer now.
 - o Caption typo: “componet”.
 - o “...at $t/T = 0.25$...”. This prompted a thought: if the *stroke* is regarded as the smallest unit of repeating motion, is the use of *wingbeat* period T actually relevant or correct? Please also revisit **Figure 5**, whose plots present averaged data from single strokes (cf. **p. 10, line 245**) across $t/T = [0,0.5]$. In these plots, the time clock effectively ‘resets’ at $t/T = 0.5$ with the beginning of a new stroke, not a new wingbeat, so T may not be the best denominator. Perhaps define and use a *stroke period* instead.
 - o Please convert the colour bar labels to vector text.
 - o **Response-Figure 4** certainly indicates that the upflow is lower in magnitude and of much lesser extent than the downflow. Still, it is important at least to acknowledge the presence of upflow (the **Figure 6** caption also says *vertical* flow, which includes both directions). I suggest either (i) addressing upflow in the text/caption and clarifying that the colour bar is one-sided, or (ii) a new colour bar with one colour for upflow and another for down, blended together via white at zero, *e.g.*,

- **[line 371]** “While for...”. Please check this sentence for readability and errors.
- **[line 422 onwards]** I appreciate the addition of the Q -criterion analysis in response to another reviewer’s comment. However, it is unusual to open a section with an analysis only to consign the results to the SM. This should be addressed somehow, perhaps by reordering the section or by presenting the Q -criterion results.
- **[line 423]** I suggest finding the original literature reference for the Q -criterion.
- **[line 445]** The tenses got mixed up here. Please check for other instances of this.

- **[Figure 8]**
 - As in **Figure 6**, please show a coarser field of larger vectors.
 - Please convert the colour bar labels to vector text.
- **[Figure 9]**
 - “Mean C_L Curve, σ Area...”. Please rephrase this more conventionally and remove capitalisations as needed.
 - The mean (thick) lines get lost amid the others. Please consider plotting them above the rest in a different colour and/or line style.
 - Is there any reason why the tone of blue differs in (d)?

Conclusion

- This section needs attention. I suggest phrasing the points more succinctly.
 - Perhaps it should be *Conclusions* (plural) because there is more than one.
 - Some people will only read the **Abstract** and **Conclusion**. Quantities such as J_{Vert} and Θ should therefore be presented with care.
 - I think the final paragraph detracts from the conclusions and should be deleted.

Supplementary Material

- How was the velocity gradient tensor calculated for the Q -criterion?
- **[Figure S1]** The caption says “instantaneous”. Are these not phase-averaged flow data?
- **[Figure S2]** Please check the caption for typos.

Appendix C

Response to reviewers

Our sincerest thanks to the editor and reviewers for insightful comments on our paper. All specific points have been addressed (listed in detail below) and incorporated into the manuscript revisions. Point-by-point response to the reviewers' comments have been highlighted by blue. Corrections have been coloured in red in the revised version of the manuscript. We believe these revisions have greatly improved the quality of the manuscript.

Reviewer: 1

1. I am unclear about how the authors define gust. On one hand, they are motivated by gusty winds but on the other hand, they reply that "majority of the perturbations encountered in realistic environments are likely to be of timescales much lower than those of the wingbeat..." therefore, they assume "wing flapping frequency and simulated as a uniform constant velocity perturbation in the freestream flow". Therefore, in this research gust is not a factor? Why mention it or discuss it? Why not simply remove the entire discussion on gust as it is irrelevant to the studied topic.

We would like to take this opportunity to distinguish the definition of "gust" from "inflow". In this context, interaction between flapping wings and freestream disturbances may be characterised as either fast or slow, depending on the temporal profile of the disturbance in relation to the wingbeat. Fast disturbances (or "gusts") are those where the duration over which the wing experiences a significant portion of the disturbance magnitude (as a fraction of the wing velocity) is of the same order or smaller than the timescale of their wingbeat. While slow disturbances (or "inflows") may be considered as those with frequencies of variation much lower than the wing flapping frequency. The focus of this study was on simulating slow perturbations (or "inflows") where the spatial extent of the disturbance would also likely be large in comparison to the size of the wing, thus a reasonable simulation of the disturbance can be obtained by plunging the wing at constant velocity in quiescent fluid. Please note that term "inflow" has been used in the text and title.

2. PIV: How do you decide what is "significant variation between image pairs for each phase were not noted"? Was this a visualize inspection or a statistical test. 10 image pairs seems to be a very low number to enable capturing any dynamic phenomena associated with fluid mechanics, specifically the formation of LEV.
 - o Though it is indeed desirable to have taken several image pairs, in this study we took only 10 image pairs for the different conditions. The authors acknowledge that this is a potential limitation of the study. To further check the sensitivity of 10 image pairs and test statistically stationarity we captured the vorticity fields using phase averaging over 5 random image pairs. Comparison between the vector fields obtaining from 10 and 5 image pairs demonstrated nominally similar flow behaviour (Figure 1 (a) and (b)) with less than 5% difference (Figure 1 (c)).

Figure 1. Phase averaged vorticity field based on (a) 10 DPIV image pairs. (b) 5 DPIV image pairs at mid-stroke and $J_{vert} = 0.4$. (c) Mean standard deviation calculated by $\bar{\sigma} = \sqrt{\frac{\sum_{i=1}^2 (\omega(i) - \bar{\omega})^2}{2}} / \bar{\omega}$.

Moreover, “This sample size was deemed sufficient ...” [p. 11 , line 269 in the original version] was removed from the text and the following sentences were added to the captions of Figures 6 and 8:

Figure 6. **Vertical component of the flow phase averaged over 10 DPIV image pairs.** [line 354]

Figure 8. **Phase averaged vorticity field based on 10 DPIV image pairs at mid-stroke.** [line 446]

3. PIV setup: 4 mm light sheet seems a bit large that can cause significant out of plane motion for the particles while moving within the light sheet, was this issue recognized?

We acknowledge the presence of out of plane motion. However, this motion is likely insignificant compared to the in-plane motion. The framerate was also maximized to reduce out-of-plane displacement. The following sentence was added to the text:

Due to the presence of strong chordwise flow most of the particle motion was along the illuminated plane and effects of out of plane motion was minimized by reducing image exposure during frame acquisition. [line 256 to 258]

4. Wing-wake interaction: It is still unclear how the authors based on visualize inspection of the velocity field determines whether interaction occurs or not. Interaction can occur within the velocity gradients; for example between the vorticity and stress fields (see Tsinober A, 2001, Introduction to Turbulence). A statistical analysis that provide some insight into the skewness of the interaction between the velocity gradients may indicate the existence of such interaction is only one option out of many procedures that provide a quantitative observation.

The authors appreciate the reviewer’s feedback. Although we agree that the existence of wing-wake interaction can be quantified by a statistical analysis of the velocity gradient, we believe that the presence of the downward flow region around the wing in the velocity field can indicate likely existence of wake interaction. We would like to take this opportunity to summarize determination of wing-wake interaction based on the velocity field:

In the quiescent case wake is normally shed beneath the wing. When an upwards inflow is induced the wake is translated upwards and into the path of the reciprocal stroke leading to creation of a large downwards flow region around the wing. While for downwards inflow, the wake is transferred downwards further downstream from the wing resulting in no dominant downwards flow region. Therefore, the exitance of the downwards flow region ahead of the wing highlights the presence of the wake from previous stroke. Whenever, the flow ahead of the wing is dominated by the downwards flow from the wake, the effective angle of attack (Θ_{eff}) deviates from the theoretical angle of attack (Θ_{th}) resulting in perturbation in the lift and drag coefficients (for $0.3 \leq J_{vert} < 0.7$). In contrast, in the absence of a dominant downwards flow around the wing (downwards inflows and upwards inflows $0.7 \leq J_{vert} \leq 1$) the effective angle of attack matches the theoretical angle of attack and \bar{C}_L and \bar{C}_D follow the nominally linear trend, traits which verify the absence of a pronounced wing-wake interaction.

Reviewer: 2

Introduction

1. The new opening sentences position the study better. But I suggest rewording the first sentence such that “their” refers grammatically to “insects” and not “agility and manoeuvrability”, even if the intended meaning is clear from the context.

The sentence was revised as follow.

Remarkable agility and manoeuvrability of insects **have led to an increasing interest in insect-inspired flight mechanisms for small-scaled flying vehicles.** [line 37 to 38]

2. The new wording implies that there exist other categories of study beyond *numerical* and *experimental*. Please rephrase accordingly.

The sentence was revised as follow.

... most studies available in the literature have investigated the effects of frontally and laterally oriented inflows using numerical [7-9] and experimental [10-13] techniques. [line 65 to 66]

3. "...in two-wing miniature flying vehicles." This seems somewhat incongruous. I suggest moving it up or just deleting it.
We have taken the suggestion to delete "leading to a substantial rolling moment in two-wing miniature flying vehicles".

Experimental set-up and procedure

4. This figure is clearer than before.
 - While I appreciate the left-hand image probably originated in CAD and a full vector rendition may not be possible, please ensure the final resolution of the raster elements is high.
We have taken the suggestion and increased the resolution of Figure 1 (a). [p.5, Figure 1]
 - Please consider homogenising the appearance of the text and arrows across the figure, perhaps converting them all to vector format. Dimension units may not be needed on the left-hand side if the caption has them.
The format of text and arrows
The format of the text and arrows in figure 1 (a) was converted to vector format. In addition, dimension units were removed in Figure 1 (a). [p.5, Figure 1]
5. "The wing was...". This sentence could still explain that the force coefficients level off after $h/c \geq 1.8$, even if **response-Figure 1** is omitted from the manuscript. Perhaps that figure could go in the Supplementary Material (SM) instead.
While we appreciate the reviewer's feedback, we respectfully decline the suggestion. **Response-Figure 1** cannot be included in the Supplementary Material, because another publication is being planned on those results.
6. [Figure 2]
 - Presumably "RAD" (y-axis) denotes *radians*? May I suggest *degrees* instead? Indeed, both ϕ and θ have units of *degrees* in the nomenclature.
We have taken the suggestion and used degree for y-axis in Figure 2 (a). [p.7, Figure 2]
 - The GoPro data are instructive; the stroke motion profile in **response-Figure 3** looks fairly sinusoidal. Do these data pertain to zero inflow? Might vertical translation of the flapper affect the stroke and/or pitch kinematics?
Yes, the data were collected at $J_{vert} = 0$. Moreover, the vertical translation of the flapper did not affect the wing kinematics.
7. "A J_{vert} of up to and including 1..." again sounds a bit clunky. This sentence might be deleted altogether.
The following sentence was deleted. "A J_{vert} of up to and including 1 enables investigation over a large range of potential inflow perturbations which flapping wing flyers could realistically experience."
8. [Figure 3]
 - If this cannot made vector, please increase the resolution.
Figure 3 was generated again with a higher resolution. [p.9, Figure 3]
 - The text labels are a bit small.
Font size of the text labels was increased.

9. "...phase-averaged forces...". Apologies if I missed this, but how many strokes were used to phase-average the forces? Presumably 10, as for PIV? If missing, please add this information somewhere.

We thank the reviewer for pointing out this issue. The text was revised as follow:

The data was phase averaged **over 20 strokes** and filtered using a low-pass, 4th order,... [line 226]

10. "Results were analysed...". Perhaps this should move up the paragraph for better ordering.

We thank the reviewer for pointing out this issue. The sentence was moved up. [line 262 to 263]

11. "This sample size was deemed sufficient as significant variation between image pairs for each phase were not noted.". This implies that the vector fields were inspected visually, which alone would be insufficient justification for deciding the number of PIV vector fields. But there is perhaps a more fundamental issue: the results figures (notably **Figure 9**) later show that the time-resolved force profile varies between strokes, particularly for inflows in the R_2 regime. This means the flow field varies as well—often in turbulent fashion (**line 464**). Are just 10 PIV realisations enough to provide robust statistical estimates of the mean flow in turbulent conditions?

Though it is indeed desirable to have taken several image pairs, in this study we took only 10 image pairs for the different conditions. The authors acknowledge that this is a potential limitation of the study and we need to have statistical stationarity. To further check the sensitivity of 10 image pairs we captured the vorticity fields using phase averaging over 5 random image pairs. Comparison between the vector fields obtaining from 10 and 5 image pairs demonstrated the nominally similar flow behaviour (Figure 1 (a) and (b)) with less than 5% difference (Figure 1 (c)).

- o Moreover, the sentence : "This sample size was deemed sufficient ..." [p. 11 , line 269 in the original version] was removed and the following sentence was added to the captions of Figures 6 and 8:

Figure 6. **Vertical component of the flow phase averaged over 10 DPIV image pairs.** [line 354]

Figure 8. **Phase averaged vorticity field based on 10 DPIV image pairs at mid-stroke.** [line 446]

Results and discussion

12. "...decreased rapidly during pronation...". This seems like the same issue as before (concerning $-J_{vert}$). Please check this and other instances carefully again.

The text was revised as followed:

...decreased rapidly during pronation **when J_{vert} was changed from 0 to -0.3.** [line 282]

13. "...instantaneous...". Not phase-averaged?

The sentence was revised as follow:

... the profile of the **phase-averaged time traces of C_L and C_D** matched those measured in quiescent conditions. [line 309]

14. Please change "however" to "but".

"however" was changed to "but". [line 331]

15. [**Figure 5**] This figure is now easier to read, but please address the caption—it mentions both *phase-averaged* and *instantaneous*, which strictly are different.

"instantaneous" was deleted.

16. "...appears to be translated upwards...". This flow feature is at the crux of the study. Is it not possible to be more definitive?

While we appreciate the reviewer's feedback, we believe that the translation of the wakes during different strokes has been clarified in Section 3.2.1. Therefore, no changes were made in the text.

17. [line 346] "...downward flow region...". With my previous comment about reference frames, I was referring to the reference frame of the *vector fields* in **Figure 6**, not the panels in the figure. Nonetheless, the addition of roman numerals is helpful.

While we appreciate the reviewer's feedback, we believe that referring to the reference panels (roman numerals) clarifies sufficiently the flow fields for different inflow conditions in the text. Therefore, no changes were made in the text.

18. [Figure 6] The vector fields are clearer now.

○ Caption typo: "componet".

The typo was fixed.

○ "...at $t/T=0.25$...". This prompted a thought: if the *stroke* is regarded as the smallest unit of repeating motion, is the use of *wingbeat* period T actually relevant or correct? Please also revisit **Figure 5**, whose plots present averaged data from single strokes (cf. p. 10, line 245) across $t/T=[0,0.5]$. In these plots, the time clock effectively 'resets' at $t/T=0.5$ with the beginning of a new stroke, not a new wingbeat, so T may not be the best denominator. Perhaps define and use a *stroke period* instead.

While we appreciate the reviewer's feedback, we respectfully disagree. The definitions of the wingbeat and stroke have been already clarified in Section 2.2. We believe that introducing a new parameter for time such as *stroke period* makes the text more confusing. Thus, no changes were made in the text.

○ Please convert the colour bar labels to vector text.

Format of the colour bar labels was changed.

○ **Response-Figure 4** certainly indicates that the upflow is lower in magnitude and of much lesser extent than the downflow. Still, it is important at least to acknowledge the presence of upflow (the **Figure 6** caption also says *vertical* flow, which includes both directions). I suggest either (i) addressing upflow in the text/caption and clarifying that the colour bar is one-sided, or (ii) a new colour bar with one colour for upflow and another for down, blended together via white at zero, e.g.,

We have taken the suggestion and amended the caption as advised.

19. "While for...". Please check this sentence for readability and errors.

The sentence was rephrased as follow:

However, the circulation effects suggested in the previous studies were diminished for downwards inflow because the wake created during stroke reversal was likely translated well below the stroke path. As a result, the lift and drag during the first rotation phase of the stroke experienced a reduction. [line 368 to 372]

20. I appreciate the addition of the Q -criterion analysis in response to another reviewer's comment. However, it is unusual to open a section with an analysis only to consign the results to the SM. This should be addressed somehow, perhaps by reordering the section or by presenting the Q -criterion results.

Section 3.2.3 was reordered such that the Q -criterion analysis paragraph was moved down for better ordering.

21. I suggest finding the original literature reference for the Q -criterion.

A new reference [32] was added to refer to the original literature reference for the Q -criterion. [line 423]

22. The tenses got mixed up here. Please check for other instances of this.

We thank the reviewer for pointing out this error. The tenses were fixed. [line 442]

23. [Figure 8]

○ As in **Figure 6**, please show a coarser field of larger vectors.

The arrows were removed from Figure 8.

- Please convert the colour bar labels to vector text.
Format of the colour bar labels was changed.
24. [Figure 9]
- “Mean C_L Curve, σ Area...”. Please rephrase this more conventionally and remove capitalisations as needed.
The caption was revised as follow:
Stroke-by-stroke lift variation. Phase-averaged C_L , time-resolved standard deviation σ , and individual stroke C_L curves for ... [line 463]
 - The mean (thick) lines get lost amid the others. Please consider plotting them above the rest in a different colour and/or line style.
The colour and style of the individual stroke C_L curves were changed for better distinguishing the phase-averaged C_L (thick line) from the individual stroke C_L . [p.20, Figure 9]
 - Is there any reason why the tone of blue differs in (d)?
The same tone of blue was used for all figures.

Conclusion

25. This section needs attention. I suggest phrasing the points more succinctly.
- Perhaps it should be Conclusions (plural) because there is more than one.
“Conclusion” was changed to “Conclusions”.
 - Some people will only read the Abstract and Conclusion. Quantities such as J_{Vert} and Θ should therefore be presented with care.
 J_{Vert} has been already defined in the Abstract and Conclusions. Moreover, definition of Θ was added to the Conclusions.
The effective angle of attack (Θ_{eff} , determined from the velocity vector arrays obtained from DPIV images) was found to be strongly affected by ... , the theoretical angle of attack (Θ_{th} , calculated based on the known velocity of the wing and inflow) resulting in....[line 485 to 489]
 - I think the final paragraph detracts from the conclusions and should be deleted.
The final paragraph was deleted.

Supplementary Material

26. How was the velocity gradient tensor calculated for the Q -criterion?
The velocity gradient tensor was calculated through Equation (1).

$$\nabla \mathbf{v} = \begin{bmatrix} \frac{\partial w}{\partial x} & \frac{\partial w}{\partial y} \\ \frac{\partial v}{\partial x} & \frac{\partial v}{\partial y} \end{bmatrix}, (\nabla \mathbf{v})^T = \begin{bmatrix} \frac{\partial w}{\partial x} & \frac{\partial v}{\partial x} \\ \frac{\partial w}{\partial y} & \frac{\partial v}{\partial y} \end{bmatrix} \quad (1)$$

27. [Figure S1] The caption says “instantaneous”. Are these not phase-averaged flow data?

These are phase-averaged flow data. The caption was changed as follow:
Vortex identification. Phase-averaged Q -criterion at ...

28. [Figure S2] Please check the caption for typos.
The caption was revised as follow:
Variation in force production per stroke. Evolution of $\bar{\sigma}/\bar{C}_L$ and $\bar{\sigma}/\bar{C}_D$. The mean standard deviation ($\bar{\sigma}$) was obtained by time-averaging the phase-averaged σ under each J_{Vert} .

Appendix D

Review of manuscript for R. Soc. Open Sci.

Effects of Uniform Vertical Inflow Perturbations on the Performance of Flapping Wings

by Mazharmanesh *et al.*

In summary: the manuscript has improved. I must restate, though, my nagging concern about the low number of data samples. This issue should be addressed in the manuscript with, for example, mention of the limitations it places on how the data can be used and the conclusions that can be drawn from them. This seems possible—I offer suggestions below.

Please accept advanced apologies for any errors, misinterpretations or omissions in my review.

General points

- Please check new figures for neatness, including text and line placement. Also, please check the consistency of label capitalisations throughout.
- Please check the reference list for formatting discrepancies. I appreciate that the job of final formatting may belong to the journal.

Specific points

Experimental set-up and procedure

- **[line 166]** I still suggest changing the ending. Perhaps “...at the top of the experiment tank, where surface effects were negligible.” At least, please swap “minimise” for “reduce”.
- **[line 176]** “Data was...”. Please change this to “Data were...”. There are other instances of this—please change them accordingly.
- **[Equations 3 and 4]** Both “×” and “*” can be removed.
- **[line 256]**
 - Please change this to “...the effects of out-of-plane motion were...”.
 - Have out-of-plane effects truly been “minimized”? I strongly suggest “lessened” instead (or “reduced”, but this would clash with “by reducing”).
- **[line 257]** “...until acceptable signal to noise...”. I suggest “...still ensuring an acceptable signal-to-noise ratio.”. The SNR could even be given.
- **[line 265]** The visualisations supplied to Reviewer 1 are instructive, but the very low PIV sample size ($N = 10$) is still concerning. At this stage, it is best simply to acknowledge this limitation in the manuscript. Some points that might be mentioned:
 - Sample size is less important for first-order statistics, including the mean (convenient, as the main PIV figure shows mean flow). This *may* help: <https://doi.org/10.1016/j.flowmeasinst.2012.04.013>
 - The vector and vorticity fields are used in a qualitative way, *i.e.*, for making observations about patterns in the flow, rather than any deep quantitative analysis. Large sample sizes and accurate statistics are therefore less important. **Figure 7** is the only one (if I recall correctly) that relies on numerical data; and that uses a column-average, which is presumably better. Still, I would consider rounding PIV-derived flow angles to zero decimal places (**line 405 onwards**).
 - There are obvious trends in the data, even at this low sample size (as in the response to Reviewer 1).

To some extent, this sample-size issue also applies to the force data ($N = 20$). Limitations should be acknowledged in that case, too.

Results and discussion

- **[line 281]** “...was changed...”. Perhaps “...as J_{vert} was varied...”?
- **[line 328]** “phase averaged” is missing its hyphen. In other places, it has it—please check this throughout for consistency.
- **[line 341]** My original point was that the word “appears” might convey uncertainty about this important result.
- **[Figure 6]** On my previous point: *stroke period* (defined early and used exclusively) might be more relevant because only individual strokes are considered in the data—not full wingbeats. As such, the range $0.5 < t/T < 1$ does not exist. Not worth considering?
- **[section 3.2.3]** I feel this section needs work.
 - **[Line 421]** This is the first reference to **Figure 8**. I suggest “shows”, as opposed to “showed”.
 - The omission of the Q-criterion results still seems unusual to me. Why not include some of them in **Figure 8**? As it stands, the text refers to **Figure 8**, then **Figure S1**, then **Figure 8** again. Overall, the Q-criterion results do not seem to be discussed with the necessary conviction.
 - **[line 435]** Might “...smaller LEV forming...” be better?

Supplementary Material

- On the Q-criterion, I was interested in how the velocity gradients were calculated from the vector field. By a differencing approach? Or did the PIV algorithm supply them?

Appendix E

Response to reviewers

Our sincerest thanks to the editor and reviewers for comments on our paper. All specific points have been addressed (listed in detail below) and incorporated into the manuscript revisions. Point-by-point response to the reviewers' comments have been highlighted by blue. Corrections have been coloured in red in the revised version of the manuscript.

Reviewer: 1

My concerns has been answered partially because 10 image pairs are not sufficient to deduce any meaningful insight on the fluid dynamics of the phenomena. Yet, I do understand the complexity of performing new experiments. Perhaps, a clear statement by the authors about the validity of their observation that is sensitive to the lack of statistical convergence.

We agree that PIV sample size ($N=10$) is a potential limitation of the study and acknowledge this limitation in the manuscript. However, we believe that there are obvious trends in the data, even at this low sample size.

The following sentences were added to the text and sample size has been explicitly mentioned in the caption of figures 6 and 9:

Despite the low PIV sample size ($N=10$), key flow features commonly noted on flapping wings such as LEVs, were clearly discernible, however further detailed measurements of the flow profile will be beneficial. [line 267 to 269]

Reviewer: 2

Experimental set-up and procedure

1. I still suggest changing the ending. Perhaps "...at the top of the experiment tank, where surface effects were negligible." At least, please swap "minimise" for "reduce".
We have taken the suggestion to change the ending. The sentence was revised as follow:
"...at the top of the experiment tank, where surface effects were negligible." [line 168]
2. "Data was...". Please change this to "Data were...". There are other instances of this—please change them accordingly.
We thank the reviewer for pointing out this issue. "Data was..." was replaced by "Data were..." throughout the manuscript. [line 176], [line 198], and [line 226]
3. [Equations 3 and 4] Both "x" and "*" can be removed.
"x" and "*" were removed from Equations 3 and 4.
4.
 - o Please change this to "...the effects of out-of-plane motion were...".
We thank the reviewer for pointing out this issue. The sentence was revised as follow:
"...and the effects of out of plane motion were lessened ..." [line 257]

- Have out-of-plane effects truly been “minimized”? I strongly suggest “lessened” instead (or “reduced”, but this would clash with “by reducing”).
“minimized” was replaced by “lessened”.
“...and the effects of out of plane motion were lessened by reducing exposure during frame acquisition still ensuring an acceptable signal-to-noise ratio.” [line 257]
- 5. “...until acceptable signal to noise...”. I suggest “...still ensuring an acceptable signal-to-noise ratio.”. The SNR could even be given.
We have taken the suggestion to revise the sentence as above. [line 258]
- 6. The visualisations supplied to Reviewer 1 are instructive, but the very low PIV sample size ($N=10$) is still concerning. At this stage, it is best simply to acknowledge this limitation in the manuscript. Some points that might be mentioned:
 - Sample size is less important for first-order statistics, including the mean (convenient, as the main PIV figure shows mean flow).
 - The vector and vorticity fields are used in a qualitative way, *i.e.*, for making observations about patterns in the flow, rather than any deep quantitative analysis. Large sample sizes and accurate statistics are therefore less important. **Figure 7** is the only one (if I recall correctly) that relies on numerical data; and that uses a column-average, which is presumably better. Still, I would consider rounding PIV-derived flow angles to zero decimal places (**line 405 onwards**).
 - There are obvious trends in the data, even at this low simple size (as in the response to Reviewer 1).

The authors would like to thank the reviewer for this advice. We have taken this suggestion to address the low PIV sample size as a limitation in this study. The following sentences were added to the text and sample size has been explicitly mentioned in the caption of figures 6 and 9:

Despite the low PIV sample size ($N=10$), key flow features commonly noted on flapping wings such as LEVs, were clearly discernible, however further detailed measurements of the flow profile will be beneficial. [line 267 to 269]

To some extent, this sample-size issue also applies to the force data ($N=20$). Limitations should be acknowledged in that case, too.

While we appreciate the reviewer’s feedback, we respectfully disagree. The standard deviation in Figure S2 (original version) demonstrates that the levels of uncertainty in the force measurements are sufficiently low, thus the sample size is large enough for force data.

Results and discussion

- 7. “...was changed...”. Perhaps “...as J_{Vert} was varied...”?
The sentence was changed as follow:
“... decreased rapidly during pronation as J_{Vert} was varied from 0 to -0.3” [line 284]
- 8. “phase averaged” is missing its hyphen. In other places, it has it—please check this throughout for consistency.
We thank the reviewer for pointing out this issue. “phase averaged” was changed to “phase-averaged”. [line 330]
- 9. My original point was that the word “appears” might convey uncertainty about this important result.
We have taken the suggestion to revise the sentence as follow:
“... the wake, which is normally shed beneath the wing, is translated upwards and into the path of the reciprocal stroke...”. [line 343]

10. On my previous point: *stroke period* (defined early and used exclusively) might be more relevant because only individual strokes are considered in the data—not full wingbeats. As such, the range $0.5 < tT < 1$ does not exist. Not worth considering?

While we appreciate the reviewer's feedback, we believe that wingbeat frequency has been commonly used in the literature to normalize time. To generalize time scaling and make it less confusing, T is used as the denominator.

11. [section 3.2.3] I feel this section needs work.

- [Line 421] This is the first reference to **Figure 8**. I suggest “shows”, as opposed to “showed”. The sentence was revised as follow:
“...shows characteristics of the presence of a LEV...” [line 435]
- The omission of the Q-criterion results still seems unusual to me. Why not include some of them in **Figure 8**? As it stands, the text refers to **Figure 8**, then **Figure S1**, then **Figure 8** again. Overall, the Q-criterion results do not seem to be discussed with the necessary conviction.

We have taken the suggestion to move Figure S1 from the Supplementary Material to the main text. We also reordered sentences in the first paragraph of Section 3.2.3. The following sentences were added to the text:

The anti-symmetric part ($[\Omega]$) and symmetric part ($[\psi]$) of the velocity gradient tensor were calculated for every point within each 52×52 - point velocity vector field from DPIV image pairs. Figure 8 exhibits $Q > 0$ in all inflow ratios which indicates the presence of LEV. [line 426 to 429]

Figure 1. Vortex identification. Phase-averaged Q-criterion at $t/T = 0.25$ normalized with R_{RoG} and U_{RoG} under (a) upward inflow for *i, ii, vii* and *viii* in the regime R_1 and *iii – vi* in the regime R_2 . (b) downward inflow. [line 431]

- Might “...smaller LEV forming...” be better?
We thank the reviewer for this suggestion. The sentence was revised as follow:
“...result in smaller LEV forming over the wing...”. [line 440]

Supplementary Material

12. On the Q-criterion, I was interested in how the velocity gradients were calculated from the vector field. By a differencing approach? Or did the PIV algorithm supply them?
The velocity gradients were calculated by a differencing approach.